

**20th-century ecological disasters in central European monoculture pine**
**plantations led to critical transitions in peatlands**
Mariusz Bąk[1], Mariusz Lamentowicz[1], Piotr Kołaczek[1], Daria Wochal[1], Michał Jakubowicz[2],
Luke Andrews[3], Katarzyna Marcisz[1]
[1]Climate Change Ecology Research Unit, Faculty of Geographical and Geological Sciences,
Adam Mickiewicz University, Poznań, Poland
[2]Isotope Research Unit, Faculty of Geographical and Geological Sciences, Adam Mickiewicz
University, Poznań, Poland
[3]School of Biological and Environmental Sciences, Liverpool John Moores University,
Liverpool, United Kingdom
*Correspondence to*: Mariusz Bąk, mariusz.bak@amu.edu.pl

## 16 Abstract

The frequency of extreme events worldwide is steadily increasing. Therefore, it is crucial to
recognize the accompanying response of different ecosystems. Monoculture tree plantations
with simplified ecosystem linkages are particularly vulnerable to catastrophic events like fires,
wind throws, droughts and insect outbreaks. These events threaten forests and other associated
ecosystems, including peatlands, which are extremely important in regulating the global carbon
cycle and thus mitigating the effects of a warming climate. Here, we examined how a peatland
in one of Poland's largest pine plantation complexes responded to some of the largest
environmental disasters observed in the 20th century across Central Europe – the 1922–1924
outbreak and the 1992 fire. As a disturbance proxy, we used a multi-proxy palaeoecological
analysis supported by a neodymium isotope record. We showed several critical transitions in
the peatland associated with extreme events and anthropogenic impacts, which triggered
significant changes in the peatland's ecological status.

## 30 Introduction

In recent decades, peatlands have been subjected to intense and ever-increasing climatic and
anthropogenic pressures (Zhang et al., 2022). Hydrologically unstable due to diverse
anthropogenic impacts, they are becoming extremely susceptible to various types of





disturbances and extreme phenomena, which are a threat to human health, cause economic
losses, and contribute to the amplification of the global warming effect (Kiely et al., 2021; Page
et al., 2002). Peatlands have evolved from being net $CO_2$ sinks to $CO_2$ emitters in every climate
zone – from tropical (Deshmukh et al., 2021; Page et al., 2022) to boreal realm (Ofiti et al.,
2023; Turetsky et al., 2011; Wilkinson et al., 2023). This is particularly important because
peatlands are precious ecosystems accumulating a third of the world's soil carbon stocks (Parish
et al., 2008), twice the entire biomass of the world's forests (Beaulne et al., 2021).
The danger is even higher for peatlands located within monoculture tree plantations that have
simplified linkages (Chapin et al., 2012) and thus are more sensitive to fires, strong winds,
droughts, and insect outbreaks, that are more common in recent years (Seidl et al., 2014;
Westerling, 2016). These negative impacts have been recorded for various peatlands, including
those in Central and Eastern Europe (Leonardos et al., 2024; Łucόw et al., 2021). Forests cover
31% of Poland's area, equivalent to 94,770 km$^2$ (Statistical Office in Białystok, 2023). More
than half of this forest cover comprises coniferous forests dominated by Scots pine (*Pinus*
*sylvestris* L.). It is mainly the result of planned forest management in modern-day Poland in the
19$^{th}$ and 20$^{th}$ centuries (Broda, 2000). Pine monocultures were easier to manage and grew faster
on poor soils, securing the continuous supply of raw material for the growing timber industry
(Broda, 2000).
It is essential to recognize how peatlands at different latitudes respond to a warming climate
and how they respond to changes resulting from the management of their surroundings (land
use change), including the planned forests and monoculture tree plantations. Thanks to their
anaerobic and acidic conditions, peatlands are excellent preservers of various types of micro-
and macrofossils (Rydin and Jeglum, 2013; Tobolski, 2000). Thus, they are valuable archives
of the changes occurring in the peatland (autogenic change) and its surroundings (allogenic
changes).
Multi-proxy palaeoecological studies (including analyses of several proxies, e.g., testate
amoebae, plant macrofossils, pollen, charcoal and others) are an excellent tool for
reconstructing the peatland development (Birks and Birks, 2006; Mitchell et al., 2000).
Particularly broad insight can be provided when dendrological (Bąk et al., 2024) or geochemical
methods (Fiałkiewicz-Kozieł et al., 2018; Gałka et al., 2019; Marcisz et al., 2023b) are included.
In recent years, the neodymium (Nd) isotope composition of the peat-hosted mineral matter has
been increasingly used in palaeoecological studies. Among the various applications, the method
has been used to determine distant sources of atmospheric dust (Allan et al., 2013; Fagel et al.,
2014; Pratte et al., 2017) and the signal associated with anthropogenic pollution (Fiałkiewicz-



Kozieł et al., 2016). Marcisz et al. (2023b) used this method to identify local disturbances in
peat, such as fires or deforestation.
The environmental past of the largest European forest complexes, including the Noteć Forest
area in Poland studied here, is insufficiently understood. These forests were affected by some
of the most severe environmental disasters of the 20[th] century that took place in pine-dominated
forests across Central and Eastern Europe – the 1922-1924 insect outbreak and the 1992 fire.
The only palaeoecological data documenting these events in the Noteć Forest come from the
Rzecin peatland (Barabach, 2014; Lamentowicz et al., 2015; Milecka et al., 2017). However,
not all the evidence of past dramatic events has been well preserved in the previously studied
core, leaving the question of the impact of insect outbreaks and fire on peatlands open for further
investigation. Small peatlands are usually less resilient to disturbances than large ones
(Lamentowicz et al., 2008). The changes caused by extreme events can lead a peatland to reach
a critical transition, that is, to cross a tipping point after which it does not return to its previous
hydrological and trophic conditions (Dakos et al., 2019; Lenton et al., 2008, 2019). So far,
peatland research has focused chiefly on the tipping points associated with changes in
groundwater levels due to a warming climate, fires, pollution, carbon sequestration, or opening
landscape caused by agricultural development (Fiałkiewicz-Kozieł et al., 2015; Jassey et al.,
2018; Lamentowicz et al., 2019a, b; Loisel and Bunsen, 2020). Except for these issues, there is
a need for a broader recognition of the consequences of insect outbreaks in forest areas and the
accompanying forest management.
In this article, we focus on the impact of catastrophic events on the ecosystem of the Miały
peatland in the Noteć Forest (local scale) and the broad context of such disturbances for pine
plantations in Central and Eastern Europe (regional scale). Our aims were (1) to reconstruct the
environmental history of the Miały peatland using multiproxy palaeoecological analyses
(including analyses of pollen, non-pollen palynomorphs, testate amoebae, plant macrofossils
and charcoal) and geochemical analyses (neodymium isotope signatures), and through this
reconstruction to identify peat layers corresponding to severe environmental catastrophic
events; (2) assess the impact of such disturbances on the peatland ecosystem, as well as to
understand the relation between disturbances occurring in the surrounding forest and the
peatland. We hypothesized that catastrophic events in pine plantations, including insect
outbreaks and fires, cause significant changes in the peatlands located in their area and even a
complete change in trophic and hydrological conditions, leading to a critical transition.
**Materials and methods**



## Study site

The Miały peatland is located in western Poland, about 65 km northwest of Poznań (Fig. 1). It is located within the boundaries of the Noteć Forest, one of the largest forest complexes in Poland, covering an area of about 1370 km² (Statistical Office in Białystok, 2023). The Noteć Forest is a Scots pine-dominated monoculture (*Pinus sylvestris*, 95% of the tree stand) (Sukovata, 2022). A large part of the pine forest, including our research site, is located in the 'Puszcza Notecka' protected landscape area. It is also a special protected area, 'Puszcza Notecka' (PLB300015, since 2007), and a special area of conservation, 'Dolina Miały' (PLH300042, since 2023), under Natura 2000. According to the physical-geographic regionalization, the peatland is located in the Gorzów Basin mesoregion, in the Warta and Noteć Inter-river submesoregion. It is a high glacial-alluvial terrace covered with dunes with a relative height of 20 to 40 meters (Kondracki, 2001). It has a temperate transitional climate. From 1981 to 2010, the average annual air temperature was 8.4 °C. The warmest month was July, with an average temperature of 18.8 °C, and the coolest month was January, with an average temperature of –1.1 °C. Average annual precipitation for 1981–2010 equalled 563 mm, with the maximum precipitation in July – 69 mm, and the minimum in April – 31 mm (Institute of Meteorology and Water Management, 2025).

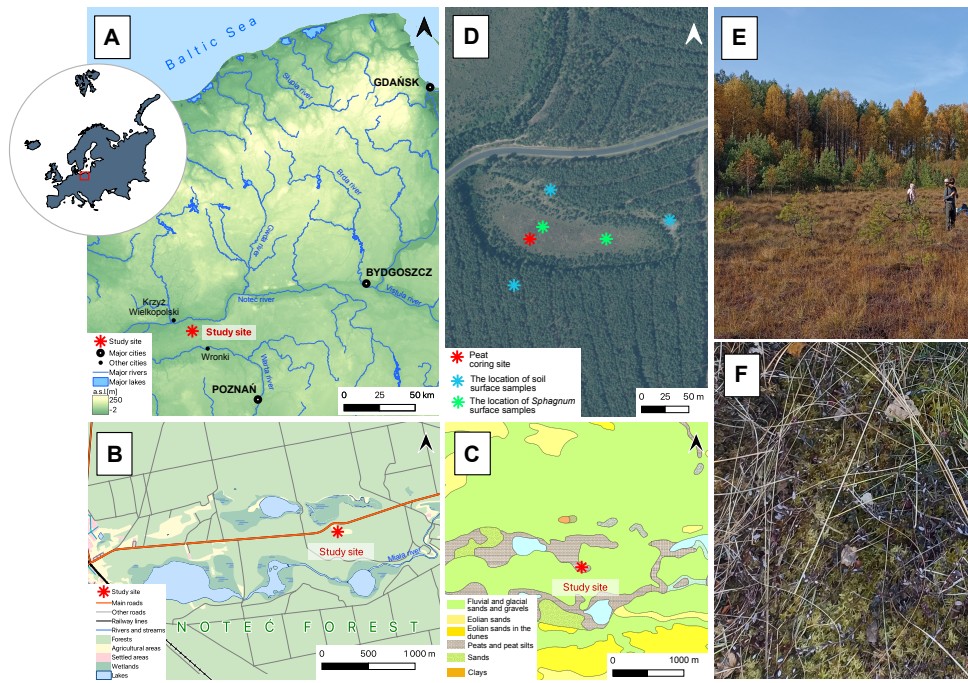



Figure 1. A-C. The location of the study site on topographic (A, B) and geological (C) maps. D. Orthophoto of the Miały peatland with sampling points (asterisks): red – peat core sampling site, blue – soil surface sampling sites for the neodymium isotope analyses, green – *Sphagnum* surface sampling sites for the neodymium isotope analyses. E. Photograph of the peatland and its forest surroundings. F. *Sphagnum* mosses covered the peatland surface.

**Fieldwork and sampling**

The peat core was collected from the western part of the peatland in October 2021 using a Wardenaar corer (chamber dimensions: 10 cm × 10 cm × 100 cm) (Wardenaar, 1987). The entire length of the sampled peat core – a 97 cm-long monolith – was analyzed. The core was subsampled continuously every 1 cm, except for the first sample (0–2 cm), which contained a living layer of peat-forming vegetation. A total of 96 samples were obtained for multi-proxy analyses, including the water content in fresh material, organic matter content in dry material, ash-free bulk density, peat accumulation rate, peat carbon accumulation rate, plant macrofossils, testate amoebae, macroscopic and microscopic charcoal, pollen and neodymium isotopes.

**Radiocarbon dating, absolute chronology and peat accumulation rates**

Ten samples containing *Sphagnum* and brown moss stems were used for accelerator mass spectroscopy (AMS) [14]C dating of the entire length of the core, conducted at the Poznan Radiocarbon Laboratory in Poland (laboratory code marked Poz; Tab. 1).

The absolute chronology of the core was based on a Bayesian age-depth model using OxCal v4.4.4 (Bronk Ramsey, 2021). The *P_Sequence* command with a parameter $k$ of 0.75 cm$^{-1}$ calculated the model, assuming $log_{10}(k/k0) = 2$, and interpolation = 1 cm. The IntCal20 (Reimer et al., 2020) and Bomb21NH1 (Hua et al., 2021) atmospheric curves were used as calibration sets. The most pronounced changes in peat composition, as manifested by changes in pollen concentration, testate amoeba species composition, and species composition of plant macrofossils, which may signal changes in peat accumulation rates, are inputted using the *Boundary* command. In this model, the *Boundary* command was input at a depth of 26 cm, with a pronounced change in pollen concentration. Two dates (laboratory code – Poz-150636 and Poz-150390) were rejected because they were after the initial modelling trajectory of the model. For better readability of the age-depth model, mean values (μ) rounded to tens were applied in the following section of the text. Peat accumulation rates were retrieved from the age-depth model using the OxCal software.



Table 1. The list of radiocarbon dates from Miały peatland with calibration. The outliers are marked with asterisks (*). The IntCal20 (Reimer et al., 2020) and Bomb21NH1 (Hua et al., 2021) atmospheric curves were used to calibrate the dates. pMC – percent modern carbon

| Laboratory code – number sample | Depth (cm) | $^{14}$C date ($^{14}$C BP) | Calibrated dates [cal. CE (2s – 95.4%) | Dated material |
|---|---|---|---|---|
| Poz-150634 | 10.5 | 114.23 ± 0.28 pMC | 1958-1962 (9.7%) 1986-1996 (85.7%) | *Sphagnum* stems |
| Poz-150451 | 20.5 | 153.88 ± 0.4 pMC | 1964-1974 (95.4%) | *Sphagnum* stems |
| Poz-150635 | 30.5 | 110 ± 30 | 1682-1738 (25.7%) 1754-1762 (1.1%) 1801-1938 (68.6%) | *Sphagnum* stems, seeds |
| Poz-150681 | 40.5 | 370 ± 40 | 1448-1530 (48.8%) 1540-1635 (46.7%) | *Sphagnum* and brown mosses stems |
| Poz-156989 | 45.5 | 750 ± 30 | 1224-1290 (95.4%) | brown mosses stems |
| Poz-150389 | 50.5 | 830 ± 30 | 1166-1269 (95.4%) | *Sphagnum* and brown mosses stems |
| Poz-156994 | 55.5 | 840 ± 30 | 1162-1266 (95.4%) | brown mosses stems |
| Poz-150636* | 60.5 | 470 ± 30 | 1407-1460 (95.4%) | *Sphagnum* and brown mosses stems |
| Poz-150390* | 70.5 | 1730 ± 30 | 248-298 (32.6%) 306-406 (62.9%) | brown mosses stems |
| Poz-156773 | 75.5 | 1595 ± 30 | 417-546 (95.4%) | brown mosses stems |
| Poz-150637 | 80.5 | 1530 ± 30 | 434-467 (11.3%) 472-519 (15.6%) 526-603 (68.6%) | *Sphagnum* and brown mosses stems, charcoal, seeds |
| Poz-150682 | 96.5 | 1910 ± 30 | 28-44 (2.9%) 58-214 (92.6%) | *Sphagnum* and brown mosses stems |

**Peat properties and peat carbon accumulation rates**

The water content in a wet sample (WC, %), organic matter content in a dry sample (ORG, %), ash content (ASH, g, %), ash-free bulk density (BD, g/cm³), peat accumulation rate (PAR, mm/yr) and peat carbon accumulation rate (PCAR, gC/m²/yr) were calculated for each of the 96 samples. For these analyses, the volume of each sample was accurately measured using calipers. Next, each sample was placed in separate crucibles, weighed, dried, and weighed again to determine the percent of WC. The dried samples were burned in a muffle furnace at 550 °C



for 5 hours and reweighed according to the protocol of Heiri et al. (2001) to determine ASH (g,
%). BD (g/cm$^3$) was calculated by dividing the weight of the dry sample by the volume of the
fresh sample and multiplied by ORG, according to Chambers et al. (2010). PAR was calculated
based on core chronology and then multiplied by the BD value obtained earlier and by 50% to
obtain PCAR, according to Loisel et al. (2014).

**Plant macrofossil analysis**

The plant macrofossils were analysed using the modified protocol of Mauquoy et al. (2010).
Each sample of approximately 5 cm$^3$ was wet sieved (mesh diameter: 200 μm). The generalized
content of the sample was estimated in percentage using a binocular microscope. Fruits, seeds,
achenes, perigynia, scales, whole preserved leaves, sporangia, and opercula were counted as
total numbers in each sample. The tissues of monocotyledon species and moss leaves (brown
and *Sphagnum* mosses) were identified on slides using a magnification of ×200 and ×400. The
material was compared with the guides (Anderberg, 1994; Berggren, 1969; Bojňanský and
Fargašová, 2007; Mauquoy and van Geel, 2007). The diagram for the analyzed proxy was
plotted using the riojaPlot package for R (Juggins, 2023).

**Testate amoeba analysis**

Peat samples for testate amoeba analysis were washed under 300 μm mesh following Booth et
al. (2010). Testate amoebae were analyzed under a light microscope with ×200 and ×400
magnifications until the sum of 100 tests per sample was reached (Payne and Mitchell, 2009);
however, in peat layers below 27 cm, the testate amoeba sums were lower (between 5 and 50)
due to the very low concentration of tests. Several keys, including taxonomic monographs
(Clarke, 2003; Mazei and Tsyganov, 2006; Meisterfeld, 2001) and online resources
(Siemensma, 2023), were used to achieve the highest possible taxonomic resolution. The results
of the testate amoeba analysis were used for the quantitative depth-to-water table (DWT) and
pH reconstructions. Both reconstructions were performed in C2 software (Juggins, 2007) using
the European training set (Amesbury et al., 2016). In layers with low testate amoeba sums, water
table reconstruction should be viewed with caution (Payne and Mitchell, 2009).

**Pollen and non-pollen palynomorphs analyses**

Samples for palynological analysis (volume: 3 cm$^3$ for 0-21 cm and 1 cm$^3$ for 21-97 cm) were
prepared using standard laboratory procedures (Berglund and Ralska-Jasiewiczowa, 1986). To



remove the carbonates, samples were treated with 10% hydrochloric acid. This step was
followed by digestion in hot 10% potassium hydroxide (to remove humic compounds) and
soaking in 40% hydrofluoric acid for 24 h (to remove the mineral fraction). Next, acetolysis
was carried out. Three *Lycopodium* tablets (Batch 280521291, containing 18,407 spores per
tablet; produced by Lund University) were added to each sample during the laboratory
procedures for the calculation of microfossil concentration (Stockmarr, 1971). Pollen, spores,
and selected non-pollen palynomorphs (NPPs) were counted under an upright microscope
(Zeiss Axio SCOPE A1) until the number of total pollen sum (TPS) grains in each sample
reached at least 500, apart from 10 samples in which pollen concentrations were very low. Two
of them (depths: 19–18 and 17–16 cm) were excluded due to extremely low pollen
concentration, and it was impossible to reach 100 grains included in TPS. Sporomorphs were
identified with the assistance of atlases, keys (Beug, 2004; Moore et al., 1991), various
publications, and the image database in the case of NPPs, for which there are no atlases (Miola,
2012; Shumilovskikh et al., 2022; Shumilovskikh and van Geel, 2020). The results of the
palynological analysis were expressed as percentages, calculations are based on the ratio of an
individual taxon to the TPS, i.e., the sum of AP (arboreal pollen) and NAP (non-arboreal
pollen), excluding aquatic and wetland plants (together with Cyperaceae and Ericaceae),
cryptogams, and fungi. A pollen diagram was drawn using the program Tilia (Grimm, 1991).

## Macro- and microcharcoal analyses

Microscopic charcoal particles (size: > 10 μm) were analyzed from the same slides as pollen
following standard protocol where the number of charcoal particles and *Lycopodium* spores
counted together exceeded 200 (Finsinger and Tinner, 2005; Tinner and Hu, 2003). Microscopic
charcoal influx or accumulation rates (MIC, particles/cm$^2$/year) were calculated by multiplying
the charcoal concentrations by peat accumulation rates (PAR) (Davis and Deevey, 1964; Tinner
and Hu, 2003).
Ninety-six contiguous samples (2 cm$^3$) were prepared for macroscopic charcoal analysis.
Bleaching was used to create a more visible contrast between the charcoal and the remaining
organic matter, following the method described by Whitlock and Larsen (2001). The samples
were sieved through a 500-μm mesh and analyzed with a binocular under ×60 magnification.
Only charcoal fragments > 600 μm were analyzed to obtain the local fire signal (Adolf et al.,
2018). Macroscopic charcoal influx or accumulation rates (MAC, particles/cm$^2$/year) were
calculated using the charcoal concentrations and PAR.



**Neodymium isotopes**

All analytical procedures and isotopic measurements were performed in the Isotope Laboratory
of the Adam Mickiewicz University, Poznań, Poland, on a Finnigan MAT 261 multi-collector
thermal ionization mass spectrometer. Details of the analytical procedures are provided by
Marcisz et al. (2023b). Peat samples, as well as surface *Sphagnum* and soil samples from both
peatlands, were dried and burned at 550 °C overnight. Prior to preparation for isotopic
measurements, the ash of peat and soil samples was dissolved on a hot plate (~100 °C for three
days) in closed PFA vials using a mixture of concentrated hydrofluoric- and nitric acids (4:1).
The ash of fresh plant material was digested in 16 N HNO$_3$. Neodymium was separated using
the miniaturized chromatographic techniques described by Pin et al. (1994) and Dopieralska
(2003). The analytical precision was monitored by analysing the USGS reference material
BHVO-2 ($^{143}$Nd/$^{144}$Nd =0.512986±0.000006 [2σ; n = 2]). Neodymium (loaded as phosphate)
was measured on Re in a double-filament configuration. Isotopic ratios were collected in a
dynamic mode. Nd isotope ratios were normalized to $^{146}$Nd/$^{144}$Nd = 0.7219. Repeated
measurements of the AMES standard yielded $^{143}$Nd/$^{144}$Nd = 0.512118 ± 10 (2σ, n = 12). Nd
isotope data are reported in the standard ε notation:

$$\varepsilon_{Nd} = \frac{\left(\frac{^{143}Nd}{^{144}Nd}\right)sample - \left(\frac{^{143}Nd}{^{144}Nd}\right)CHUR}{\left(\frac{^{143}Nd}{^{144}Nd}\right)CHUR} \times 10^4$$

where CHUR denotes the present-day Chondritic Uniform Reservoir ($^{143}$Nd/$^{144}$Nd = 0.512638
and $^{147}$Sm/$^{144}$Nd = 0.1967) (Jacobsen and Wasserburg, 1980).

**Statistical analyses**

To quantify periods of rapid botanical change and recovery, we apply the principal response
curves (PrC) to the data, as outlined by Burge et al. (2023) in their R package 'baselines'. The
multivariate palynological data (individual taxa only) was Hellinger-transformed and reduced
to a one-dimensional curve using PrC. This method is useful for detecting changes in data with
a strong underlying gradient in palaeoecological studies (Van Den Brink and Ter Braak, 1999;
De'ath, 1999). Mixed model generalised additive models (GAMMs) were then fitted to the data,
with a smoothing term accounting for temporal autocorrelation. When poor GAMM fits
occurred, adaptive splines with GAMS were compared with the GAMM to assess model fits.
Adaptive spline GAMs provide better fits to data exhibiting abrupt changes but cannot yet be



incorporated into the GAMM framework (Simpson, 2018). Periods of significant change were
identified in the GAMM models by calculating the time intervals where the confidence intervals
surrounding the first derivative did not include zero.
The phases in the palaeoecological analyses were distinguished based on changes in plant
communities obtained from palynological and plant macrofossil data.

**Results**
**Chronology, peat accumulation rates and peat properties**
The age-depth model shows the agreement index ($A_{model}$) of 61%, just above the recommended
minimum of 60% (Bronk Ramsey, 2008) (Fig. 2). The model has the highest uncertainty, with
a 95.4% confidence interval – 80 calibration years (CE) – at depths between 65.5 and 64.5 cm
(ca. 840–870 cal. CE, Fig. 2). The age of the oldest layer – 96.5 cm – was modelled at 130±45
(confidence interval: 1 σ) cal. CE (Fig. 2).

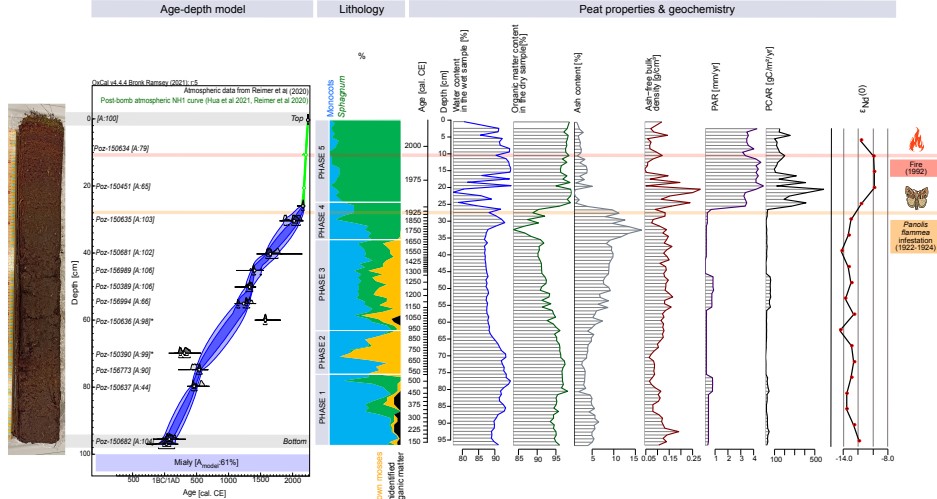


Figure 2. Bayesian age-depth model (based on $^{14}$C dating) and lithology (based on plant
macrofossils analysis) with palaeoecological phases of the peat profile in Miały (on the left
site). Changes in the physical peat properties (water content in the wet sample, organic matter
content in the dry sample, ash content, ash-free bulk density, PAR, and PCAR) and neodymium
isotope signatures – $\varepsilon_{Nd}$ – are marked. The timing of the most critical catastrophic disasters in
the 20$^{th}$ century is also marked.



The water content of the wet sample ranged from 77.0% (22–21 cm, ca. 1965 cal. CE) to 95.0%
(20–19 cm, ca. 1970 cal. CE), averaging 89.4% throughout the core (Fig. 2). Organic matter
content of the dry sample ranged from 83.6% (33–32 cm, ca. 1755–1785 cal. CE) to 99.2%
(22–21 cm, ca. 1965 cal. CE), with an average of 94.5% in the entire core (Fig. 2). Bulk density
ranged from 0.04 g/cm$^3$ (15–14 cm, ca. 1980 cal. CE) to 0.28 g/cm$^3$ (21–20 cm, ca. 1965–1970
cal. CE), with an average of 0.12 g/cm$^3$ across the core (Fig. 2). Average PAR throughout the
core was relatively slow at 1.3 mm/yr, fastest at 4.8 mm/yr (20–19 cm, ca. 1970 cal. CE),
slowest at 0.2 mm/yr (43–42 cm, ca. 1395–1440 cal. CE) (Fig. 2). The average PCAR had a
value of 73.4 gC/m$^2$/yr, the largest – 590.6 gC/m$^2$/yr (21–20 cm, ca. 1965–1970 cal. CE), the
smallest – 10.2 gC/m$^2$/yr (71–70 cm, ca. 665–700 cal. CE) (Fig. 2). Higher PAR and PCAR
values were associated with an undecomposed acrotelm zone.

**Palaeoecological analysis**
**Phase 1 (97–76 cm, ca. 130 – 520 cal. CE): very wet peatland with a dominance of**
**monocots, surrounded by mixed forest**
The local vegetation (Fig. 3) for most of this period is dominated by monocots (max. 96% of
plant macrofossil content), including *Carex*, whose achenes are found in the peat profile.
Cyperaceae pollen makes up max. 6.0% (Fig. 4). Short periods of dominance of *Sphagnum*
(max. 80%), mainly *Sphagnum* sub. *Cuspidata* (max. 40%), occur (Fig. 3). This phase is also
characterized by a high content of unidentified organic matter, reaching up to 10% (Fig. 3).
The low sums of testate amoebae do not allow for a statistically significant reconstruction of
water and pH levels in this phase (full data in the open dataset). However, among the testate
amoeba taxa, *Centropyxis aculeata* dominates quantitatively. There is a high percentage of
Cyanobacteria and algae (Zygnemataceae, *Botryococcus*) (Fig. 4) and a maximum of the
*Utricularia* curve in the pollen data (Fig. 4; max. 0.5%).
*Pinus sylvestris* (39.0–65.8%) grains are the most frequent, but the pollen of deciduous trees is
relatively common as well (Fig. 4): *Betula* (7.4–26.4%), *Alnus* (max. 17.0%), *Quercus* (max.
15.6%), *Carpinus betulus* (max. 5.8), *Corylus avellana* (max. 4.6%), *Fagus sylvatica* (max.
3.5%). Remains of *Betula* (achenes and catkin scales) are present in the plant macrofossils (Fig.

312    3).

The highest fire activity is recorded for ca. 310–330 cal. CE (macroscopic charcoal
concentration ca. 70 particles/cm$^3$, Fig. 3 and microscopic charcoal concentration ca. 420,000
particles/cm$^3$, Fig. 4) and ca. 430–455 cal. CE (90 particles/cm$^3$ of macroscopic charcoal; Fig.

316    3).






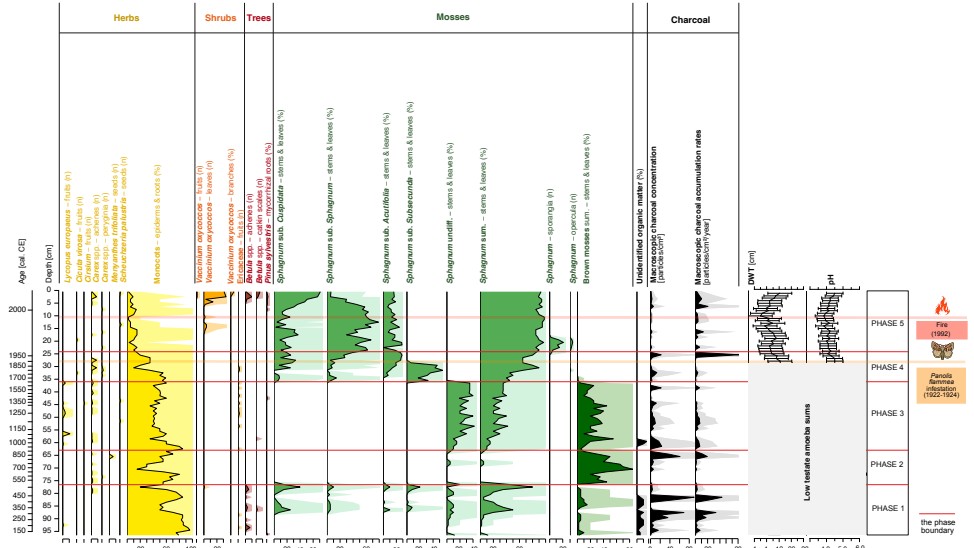


Figure 3. A diagram showing macrofossil percentages, macroscopic charcoal concentrations
and influx as a local fire proxy. Depth-to-water table and pH curves for 27–0 cm layers are also
presented. Ten times exaggeration is marked.

**Phase 2 (76–64 cm, ca. 520 – 890 cal. CE): moderately wet peatland, landscape closure –**
**increase in forestation, decrease in ruderal species**
The *Sphagnum* content decreases in favour of the brown moss (max. 85%) and monocot
remains (max. 80%), including *Carex* (achenes and perigynia of this taxon are found, Fig. 3).
Cyperaceae pollen (Fig. 4) make up between 3.4% and 8.4%. This is the only phase in which
seeds of *Menyanthes trifoliata* are found (Fig. 3), and the pollen curve maximum of this taxon
is observed (0.3%; Fig. 4).
Reconstructions of depth-to-water level and trophic conditions imply a low abundance of testate
amoebae, with a continuation of the quantitative dominance of *C. aculeata* (full data in the open
dataset). The share of freshwater bacteria and algae decreases significantly at this time (Fig. 4).
Cyanobacteria reach max. 5.9% (Fig. 4).
This period has the highest forest cover in the peatland's surroundings. Arboreal pollen accounts
for over 90% of total pollen throughout this phase (Fig. 4). Compared to the phase 1, the share
of *Betula* pollen decreases (5.1–19.1%), while the share of *Pinus sylvestris* pollen slightly
increases (44.6–65.2%) (Fig. 4). Admixture species – *Alnus* (max. 17.9%), *Quercus* (max.



9.2%), *Carpinus betulus* (max. 6.8%), *Corylus avellana* (max. 5.2%), *Fagus sylvatica* (max.
2.7%) – continue to be relatively important (Fig. 4).


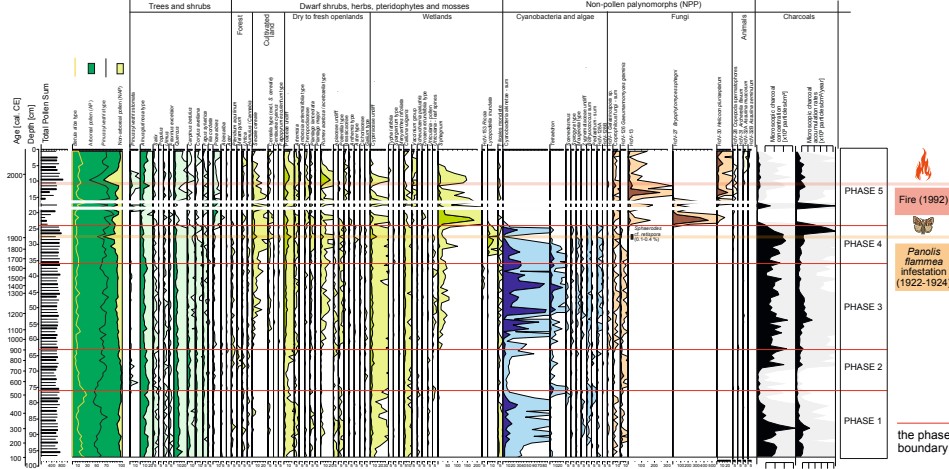


Figure 4. Pollen diagram with selected taxa presented (full list of taxa is provided in the
associated open dataset). Pollen percentages are shown in black, and 10 times exaggeration is
marked. Microscopic charcoal concentrations and influx as an extra-local fire proxy are also
presented.

Through much of the phase 2, fire activity is low. The concentration of both microscopic and
macroscopic charcoal increases markedly towards the end of this phase, reaching a maximum
of 61 particles/cm$^3$ for macroscopic charcoal (Fig. 3) and 293,600 particles/cm$^3$ for microscopic
charcoal (Fig. 4).

**Phase 3 (64–36 cm, ca. 890 – 1660 cal. CE): very wet peatland, expansion of *Sphagnum***
**mosses, development of agriculture and gradual decrease in deciduous trees**
*Sphagnum* mosses (max. 42%) appear again, although, due to the significant degree of the
material decomposition, it was not possible to determine lower taxonomic ranks in the plant
macrofossil analysis (Fig. 3). The content of the remains of monocots (max. 85%) and brown
mosses (max. 55%) remains high (Fig. 3). *Carex* achenes are also present (Fig. 3). The
percentage of Cyperaceae pollen is relatively high (2.0–7.0%; Fig. 4). This is the only phase





where fruits of *Lycopus europaeus* are found (Fig. 3). Seeds of *Scheuchzeria palustris* are also
present (Fig. 3).
The concentration of testate amoebae remains low, so again, the reconstruction of water levels
and trophic conditions should be treated with caution (full data in open dataset). Species of the
genera *Centropyxis* sp., *Cyclopyxis* sp., and *Difflugia* sp. dominate quantitatively. The increase
in Cyanobacteria (max. 82.6%) and freshwater algae, especially *Tetraëdron* (max. 24.6%) and
*Botryococcus* (max. 2.5%), is significant (Fig. 4).
The structure of the forest was relatively stable (Fig. 4). The share of arboreal pollen is high,
ranging from 86% to 94%, although with a slightly decreasing trend, compounded by declines
in admixture species. *Pinus sylvestris* represented 51-68% and *Betula* 6–15% of total pollen. At
the end of this phase, the share of *Alnus*, *Quercus*, *Carpinus betulus*, *Corylus avellana* and
*Fagus sylvatica* in total pollen is respectively: 11.6%, 5.5%, 2.0%, 1.1% and 1.6%. The declines
in the percentage of these taxa may be related to the increased contribution of Cerealia pollen
(Fig. 4). Among Cerealia, *Secale cereale* dominates, reaching a maximum of 2.2%. The
percentages of Poaceae, *Artemisia*, *Plantago lanceolata,* and *Rumex* also increase (Fig. 4).

**Phase 4 (36–24 cm, ca. 1660 – 1960 cal. CE): the further expansion of *Sphagnum* mosses,**
**an increase of *Pinus sylvestris* pollen with an episodic extreme decrease of it**
The expansion of *Sphagnum* is continued. The percentage of monocot remains decreases to
15% by the end of this phase. However, the number of achenes and perigynia of *Carex* is higher
than in any other part of the profile (Fig. 3). The percentage of Cyperaceae pollen ranges from
2.7% to 13.0% (Fig. 4). The initial part of the phase is dominated by the *Sphagnum* sub.
*Subsecunda* (Fig. 3). At the same time, *Lycopodiella inundata* appears (Fig. 4). This is the only
phase in which *Sphagnum* sub. *Subsecunda* and *Lycopodiella inundata* occur together. The
brown mosses completely disappear.
At the end of the phase 4, the abundance of testate amoebae increases (with *Galeripora*
*discoides*, *Nebela tincta,* and *Phryganella acropodia* as dominant species), which allows for
statistically significant reconstructions of the water table level and pH level (Fig. 3). The
abundance of Cyanobacteria and algae decreases distinctly; most of them disappear entirely at
the end of this phase (Fig. 4).
In the pollen dataset (Fig. 4), a further decrease in the percentage of deciduous species is
observed. In the upper part of the phase 4, the share of *Alnus*, *Quercus*, *Carpinus betulus*,
*Corylus avellana,* and *Fagus sylvatica* in total pollen is 3.4%, 1.9%, 1.2%, 1.3%, and 0.6%,
respectively. The share of *Betula* in total pollen remains at about the same level (5.9–12.2%).



A significant decrease in *Pinus sylvestris* pollen percentages and an increase in the percentages
of *Secale cereale*, Poaceae, *Plantago lanceolata*, and *Rumex* pollen occur in 1900–1926 cal.
CE.
Analysis of the macroscopic charcoal data (Fig. 3) shows one local fire event (macroscopic
charcoal concentration – 22 particles/cm$^3$, macroscopic charcoal accumulation rate – 7
particles/cm$^2$/year; 1952–1956 cal. CE). The regional fire activity (Fig. 4) remained quite high
(ca. 127,000–312,000 particles/cm$^3$ of microscopic charcoal concentration; ca. 3900–61,000
particles/cm$^2$/year of microscopic charcoal accumulation rate).
**Phase 5 (24–0 cm, ca. 1960 – 2021 cal. CE): the dominance of *Sphagnum* mosses and the**
**disappearance of Cyanobacteria and algae, the development of microscopic fungi, the**
**episodic extreme collapse of the arboreal pollen curve**
The uppermost part of the profile records further development of *Sphagnum*, initially *Sphagnum*
sub. *Sphagnum*, later *Sphagnum* sub. *Cuspidata*. The proportion of *Sphagnum* sub. *Acutifolia*
remains stable. *Sphagnum* capsule remains – sporangia and opercula – appear; we link their
presence with spores of the parasitic fungus *Bryophytomyces sphagni* (see discussion). Tree
remains (*Betula* achenes and catkin scales, *Pinus sylvestris* mycorrhizal roots) are abundant.
*Vaccinium oxyccocus* leaves appear in large numbers.
At the beginning of this phase, Cyanobacteria and algae disappear completely. Testate amoeba
species such as *G. discoides*, *Galeripora catinus*, and *N. tincta* are abundant. *G. discoides*
dominates for most of the phase 4, and the abundance of *N. tincta* increases towards its end.
The groundwater level remains constant, except for one marked fluctuation (ca. 1990–1995 cal.
CE), whereas the pH level increases gradually from ca. 1995 cal. CE (Fig. 3). Both phenomena
can be linked to the effect of the 1992 fire (see discussion).
*Pinus sylvestris* remains the dominant species in this of the profile (32.6–78.9%). Compared to
the previous phase, the percentage of *Betula* pollen increases (5.6–20.3%). One significant
decrease in the share of tree pollen, in particular *Pinus sylvestris*, is recorded in ca. 1995 cal.
CE. We interpret this as decreased forest cover after the 1992 fire (see discussion). At the same
time, a higher share of *Pinus* stomata typifies ca. 1980-2000 cal. CE layers (0.2–3.9%). We
associate this with massive needle falls associated with the fire (see discussion). *Rumex*
acetosa/acetosella type – a taxon characteristic of open and ruderal areas (Behre, 1981) –
reaches its maximum – 19.6% (ca. 1995 cal. CE), which we also interpret as an effect of the
fire. The shares of other deciduous trees – *Quercus* (max. 3.9%), *Carpinus* (max. 1.6%),
*Corylus* (max. 1.3%), *Ulmus* (max. 0.7%) decrease.



**Neodymium isotopes analysis**


The $\varepsilon_{Nd}$ values measured in the mineral matter extracted from the analyzed peat samples range
from −14.5 to −9.8. Most samples show a relatively low variability of the strongly negative Nd
isotope ratios ($\varepsilon_{Nd}$ < –12), including the most negative values in layers 61–60 and 41–40 cm.
Less negative $\varepsilon_{Nd}$ values (ranging from −9.9 to −9.8) are only observed in the upper part of the
profile, most notably in the layers 21–20, 16–15 and 11–10 cm.
Among the reference surface samples, the mineral material from the peatland surface yielded
moderately negative $\varepsilon_{Nd}$ signatures (−12.1 and −11.7), whereas the soil taken from the slopes
of the peatland catchment display strongly unradiogenic Nd isotope composition ($\varepsilon_{Nd}$ = −18.9
to −16.5; Table 2). The study site is covered by young glacial material dominated by clay and
sand derived from Scandinavia, transported and accumulated during the last glaciation (Marks,
2012). Previously, Nd isotope measurements in the young glacial sediments of another outwash
plain covered by a pine monoculture were measured only by Marcisz et al. (2023b), who
reported $\varepsilon_{Nd}$ signatures similarly negative ($\varepsilon_{Nd}$ = −26.5 to −16.6) to those in Miały.
Tab. 2. Reference $\varepsilon_{Nd}$ values measured in surface samples taken from the studied peatland and
its surrounding (1–5) and $\varepsilon_{Nd}$ values measured in peat samples.

| Nr | Sample code | $^{143}Nd/^{144}Nd_{(0)}$ | Uncertainty | $\varepsilon_{Nd}$ (t= 0) |
|---|---|---|---|---|
| 1 | MŁY01 | 0.512016 | ± 0.000011 | − 12.1 |
| 2 | MŁY02 | 0.511791 | ± 0.000010 | − 16.5 |
| 3 | MŁY03 | 0.511671 | ± 0.000012 | − 18.9 |
| 4 | MŁY04 | 0.511727 | ± 0.000012 | − 17.8 |
| 5 | MŁY05 | 0.512036 | ± 0.000011 | − 11.7 |
| 6 | MŁY5.5 | 0.512036 | ± 0.000012 | − 11.7 |
| 7 | MŁY10.5 | 0.512129 | ± 0.000010 | − 9.9 |
| 8 | MŁY15.5 | 0.512134 | ± 0.000010 | − 9.8 |
| 9 | MŁY20,5 | 0.512133 | ± 0.000009 | − 9.9 |
| 10 | MŁY25.5 | 0.512042 | ± 0.000009 | − 11.6 |
| 11 | MŁY30.5 | 0.511969 | ± 0.000010 | − 13.1 |
| 12 | MŁY35.5 | 0.511952 | ± 0.000015 | − 13.4 |
| 13 | MŁY40.5 | 0.511905 | ± 0.000010 | − 14.3 |
| 14 | MŁY45.5 | 0.511952 | ± 0.000010 | − 13.4 |
| 15 | MŁY50.5 | 0.511973 | ± 0.000010 | − 13.0 |





| 16 | MŁY55 | 0.511932 | ± 0.000010 | − 13.8 |
| 17 | MŁY60 | 0.511991 | ± 0.000010 | − 12.6 |
| 18 | MŁY65 | 0.511895 | ± 0.000017 | − 14.5 |
| 19 | MŁY70 | 0.511975 | ± 0.000008 | − 12.9 |
| 20 | MŁY75.5 | 0.511992 | ± 0.000011 | − 12.6 |
| 21 | MŁY80.5 | 0.511972 | ± 0.000010 | − 13.0 |
| 22 | MŁY85.5 | 0.511940 | ± 0.000010 | − 13.6 |
| 23 | MŁY90.5 | 0.511941 | ± 0.000010 | − 13.6 |
| 24 | MŁY95.5 | 0.511992 | ± 0.000009 | − 12.6 |
| 25 | MŁY97.5 | 0.512028 | ± 0.000012 | − 11.9 |


**Statistical analyses**

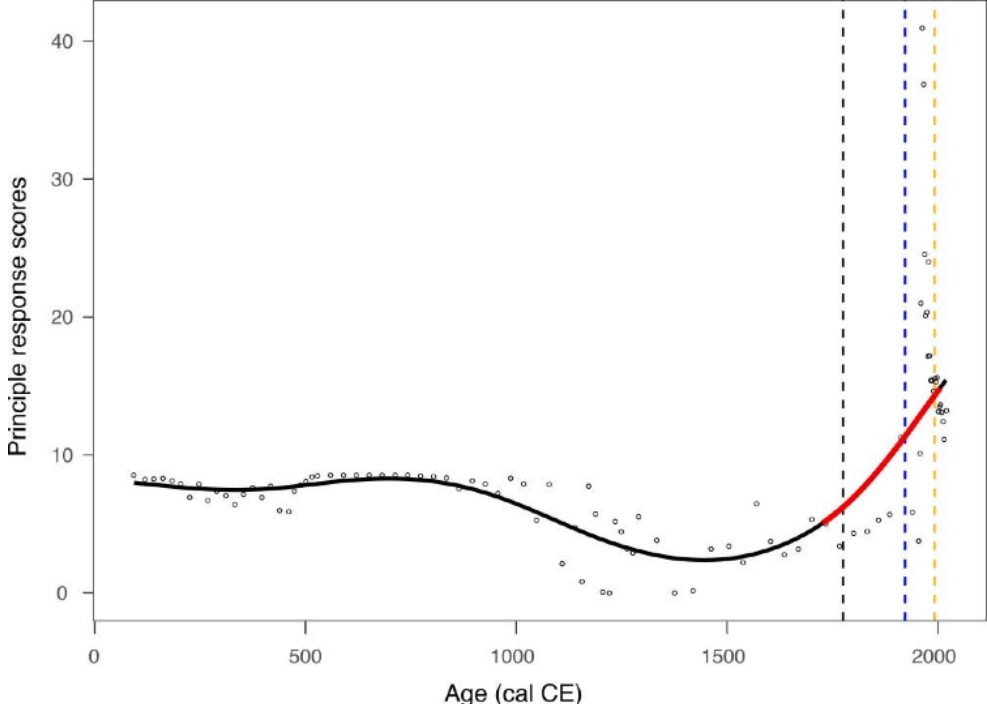


Fig. 5. Changes in the principle response curve derived from pollen count data (circles) fit with

a GAMM model fit (solid black and red lines). The red line indicates periods of rapid change.

Dashed vertical lines indicate historical periods of forest management change affecting the site:

the 1775 decree by Frederick II the Great (black); infestation by *Panolis flammea* (1922–1924;

blue), and the 1992 fire period (yellow).



The PrC explained 73% of the variance in the palynological data. However, the GAMM
provided a relatively poor fit to the data. An adaptive spline GAM provided a better explanation
of the data, with the differences between the two models primarily related to the improved fit
with the more recent samples. This suggests a possible return to previous conditions, although
these samples are more likely to be influenced by temporal autocorrelation. Despite this, the
GAMM effectively captures the general trends in the data and provides a better fit for the
earliest samples (Fig. A1). Therefore, we can proceed to use this data.
The PrC analysis revealed that changes over time occurred between the beginning of the record
and 1720 cal. CE. However, there is no substantial evidence of significant or rapid changes until
after this time. From approximately 1000 cal. CE until the 1700s, the PrC scores exhibited high
variability. A significant increase in the rate of change was identified for the period ca. 1725–
2005, as shown in Figure 5.

**Discussion**
**Combining ecological, palaeoecological, geochemical and historical data to**
**understand long-term environmental changes**
Present-day pine monoculture forests of Poland are often perceived as typical for this region by
the local populations, whereas these are highly modified forests that are significantly different
from the natural ones. Compared to natural potential vegetation maps, these areas should
possess a large proportion of deciduous taxa, e.g., oak-hornbeam (*Querco-Carpinetum*
*medioeuropaeum*) forests (Matuszkiewicz, 2008). The relatively high percentages of deciduous
tree pollen compared to the percentages of *Pinus sylvestris* pollen in historical times were
recorded at many sites from present-day pine monocultures in northern Poland (Bąk et al., 2024;
Czerwiński et al., 2021). The development of the Polish state and agriculture in the early Middle
Ages, in our data manifested by the high percentages of cereal pollen grains (incl. *Secale*
*cereale*) and taxa characteristic for open and ruderal areas (Poaceae, *Artemisia*, *Plantago*
*lanceolata,* and *Rumex*), caused a decline of deciduous species in the forest composition (Fig.
4). These changes in the forest structure were distinct but gradual; when planned management
was introduced in 18th century, however, the contribution of admixture trees started to decrease
rapidly. In 1772 CE, the area of the Noteć Forest was included in the borders of the Kingdom
of Prussia as a result of the First Partition of Poland. At that time, some of the first legal
regulations for planned forest management in the area appeared, including the 1775 CE decree
of Frederick II the Great regarding government forests in Prussia and the preference for planting



pines instead of deciduous species (Bąk et al., 2024; Jaszczak, 2008). Around this time, the PrC
analysis began to reveal periods of significant and rapid change in the palynological record.
Consequently, the forest has continued to undergo substantially rapid changes ever since, unlike
the preceding changes. The results of the PrC analysis proved to be statistically significant,
confirming the occurrence of critical transitions in the peatland on a scale that was not observed
in the older part of the core.
It is commonly assumed that outwash plains or eolian sandy dunes, remnants of the Weichselian
glaciation (to 11,700 BP), which are currently covered by extensive Scots pine monoculture in
northern Poland (e.g., the Noteć Forest, the Tuchola Forest) are not conducive to the growth of
other species and *Pinus* is a natural main forest-forming species (Magnuski, 1993; Miś, 2003).
Although pollen data suggest the domination of Pinus sylvestris since the 2nd century CE, the
distinct admixture of *Quercus*, *Carpinus betulus*, and *Corylus avellana* was recognized. The
other multi-proxy palaeoecological studies from the Noteć Forest were unable to provide such
information because the cores collected from the Rzecin peatland covered only the last 200
years and did not capture the entire background of the changes related to human activity and
subsequent forest management(Barabach, 2014; Lamentowicz et al., 2015; Milecka et al., 2017;
Słowiński et al., 2019).
The knowledge of the historical background is essential for the interpretation of the ecosystem
response to forestry practices because it enables tracing not only the composition of the forest
surrounding the peatland but also the peatlands' hydrological and trophic conditions (Bąk et al.,
2024). In this study, we recorded the presence of hydrophytes and later also helophytes and
hygrophytes (e.g., *Utricularia*, *Menyanthes trifoliata*, *Lycopus europaeus*, *Scheuchzeria*
*palustris*, *Cicuta virosa*) in the first four phases of the peatland development (up to ca. 1960
CE, Fig. 4). Combined with the high percentages of Cyanobacteria and algae (Zygnemataceae,
*Botryococcus*, *Tetraëdron*) and domination of *Centropyxis* sp., *Cyclopyxis* sp. and *Difflugia* sp.
among the testate amoebae, it indicates the existence of a shallow water body supplied not only
by rainwater and runoff but also by groundwaters (Figs. 3, 4). All these taxa disappeared in the
phase 5, after ca. 1960 CE. Monocot plants and brown mosses were displaced by the expansion
of the *Sphagnum* mosses that tolerate acid conditions. Among testate amoebae, *G. discoides*,
*N. tincta,* and *P. acropodia*, species that tolerate unstable hydrological conditions became
dominant, suggesting the lowering of the water table and substantial water table fluctuations
(Lamentowicz and Mitchell, 2005; Sullivan and Booth, 2011).
The process of peatland acidification is a natural manifestation of peatland development over
time as long as it occurs gradually. We noted a gradual transition from the moderately rich fen



to poor fen by combining *Sphagnum* sub. *Subsecunda* and *Lycopodiella inundata* taxa in the
phase 4 (ca. 1660-1960 cal. CE). However, further changes in local plant communities and
hydrological and trophic conditions toward acidification occurred abruptly, characteristic of
external interference. It can be caused by forest management, such as drainage and changes in
the forest composition (Bąk et al., 2024), including those caused by ecological disasters like
fires or insect outbreaks.
The stability of the ecosystem until the 20[th] century appears in line with the moderately variable,
unradiogenic neodymium isotope signatures of the mineral matter extracted from the peat
samples ($\varepsilon_{Nd}$ = −14.5 to −11.6). These data are similar to the results from other peatlands in the
Tuchola Forest, Poland: the Stawek peatland (−15.3 to −12.7) and Głęboczek peatland (−13.7
to −12.6) (Marcisz et al., 2023b). The notably consistent $\varepsilon_{Nd}$ values in the pre-infestation part
of the studied profile point to the dominance of local sources of the mineral matter. Strongly
unradiogenic $\varepsilon_{Nd}$ values are generally characteristic of the surface clastic sediments that
dominate the young post-glacial landscape of northern Poland (Marcisz et al., 2023a, b). The
Nd isotope signatures increased markedly after c. 1950. In their study of the Tuchola Forest
peatlands, Marcisz et al. (2023b) observed pronounced decreases in the $\varepsilon_{Nd}$ values following
major fire events, attesting to an increased supply of locally-sourced sedimentary material
favoured by the forest removal. Analogously, some decrease in the $\varepsilon_{Nd}$ values following the
1992 fire is observed in the peat profile in this study. In contrast, the deforestation following
the *Panolis flammea* infestation is followed by an increase in the Nd isotope ratios, reaching
$\varepsilon_{Nd}$ values notably higher than those observed in any of our reference samples from the peatland
catchment. Therefore, the elevated $\varepsilon_{Nd}$ values, coinciding with the notably decreased ash
contents, most likely reflect a decreased supply of the local sediments by surface runoff and
groundwater flow. This interpretation is in agreement with the acidification of the peatland; the
transition in the hydrological regime likely resulted in an increased relative role of extra-local,
aeolian sources of the sedimentary material (Allan et al., 2013; Fagel et al., 2014; Marcisz et
al., 2023a). A specific source of such $^{143}$Nd-enriched sediments cannot, however, be identified
based on the $\varepsilon_{Nd}$ record alone.
The three above-mentioned disturbance agents that influenced the status of the peatland –
anthropogenic activities connected to administrative changes, insect outbreak and catastrophic
forest fire – have all been recorded as statistically significant critical transitions in the GAMM
model (Fig. 5).



***Panolis flammea* outbreak (1922-1924) and its impact on peatland and pine plantations**

One of the most harmful documented insect outbreaks in Poland happened in 1922-1924 CE (Broda, 2003) and covered vast areas of central and eastern Europe (today's area of Germany, Poland, Lithuania, Belarus, and part of European Russia), progressing from west to east (Ziółkowski, 1924). It was caused by *Panolis flammea*, one of the most dangerous primary pests of pine trees (Szmidt, 1993). Over 500,000 hectares of forests have been defoliated in Europe (Głowacka, 2009). In the Noteć Forest, the first caterpillars found in 1921 CE did not yet herald an ecological disaster (Broda, 2003). Still, in the following two years, ca. 64,000 hectares of the forest were destroyed (Hernik, 1979). In the Potrzebowice Forest District, where our site is located, the outbreak destroyed over 90% of the forest area (~8,000 ha) (Broda, 2003).

We assume that in the pollen record, this outbreak is well recognizable (1900-1926 cal. CE; phase 4). It is marked by a sharp decrease in the percentage of *Pinus sylvestris* pollen (48.0%) compared to the neighboring layers – ca. 1875-1900 cal. CE (60.6%) and ca. 1925-1950 cal. CE (62.8%). After almost all the pine trees have been destroyed and the caterpillars had nothing to eat, they attacked the deciduous trees on which they do not usually feed (Przebieg..., 1929). In our data, a manifestation of this shift is probably the decrease in the proportion of *Betula*, *Alnus* and *Quercus* pollen. This layer also shows the highest share of Poaceae (14.7%), Cerealia (10.4%), and *Plantago lanceolata* (2.7%) pollen in the entire peat core. The share of *Rumex acetosa/acetosella* type (6.6%) is also high. The presence of taxa characteristic of open and ruderal areas indicates that the landscape has opened up due to logging activities in the destroyed forest stands. However, in the Rzecin peatland, 8 km southeast of our site, a significant decrease in *Pinus* pollen has not been observed (Barabach, 2014). According to Barabach (2014), as a result of immediate human activities, heliophytes did not develop, and a natural secondary succession did not occur at the Rzecin bog's surroundings. Barabach (2014) argued that a single pine that stands alone will produce more pollen than the same pine in a compact forest stand, referring to the individual trees that survived the disaster. Later, along with wind and water, the pollen was deposited in natural depressions, including the Rzecin peatland. However, an increase in Poaceae pollen percentages has been recorded, confirming the opening of the landscape at the Rzecin bog's surroundings.

The layers corresponding to ca. 1900-1950 cal. CE are the only portions of the core where the spores of *Sphaerodes retispora* (syn. *Microthecium retisporum*) were identified. This taxon occurs on other fungus *Tremates hirsuta*, which inhabits dead trees and their branches, as well



as recently dead and decaying wood (Bhatt et al., 2016). It mainly attacks deciduous trees, although reports from coniferous trees are known (Szwałkiewicz, 2009). Perhaps the appearance of the *S. retispora* spores in these layers reflects the presence of *T. hirsuta* on dead wood after the *P. flammea* outbreak. We also observed higher percentages of coprophilous fungi (including HdV-55A *Sordaria* type) in the layer corresponding to ca. 1900-1925 cal. CE (2.7%) compared to neighbouring layers – ca. 1875-1900 cal. CE (0.4%) and ca. 1925-1950 cal. CE (0.9%). *Sordaria* type coprophilous fungi can indicate the presence of open land and the presence of livestock, as well as wood detritus or wood burning (Lageard and Ryan, 2013; Lundqvist, 1972; Mighall et al., 2008; Wheeler et al., 2016). We point out, however, that *Sordaria* type spores can also occur on the faeces of wild herbivores and are predominantly coprophilous, meaning that this taxon may include non-coprophilous species (Shumilovskikh and van Geel, 2020). Kołaczek et al. (2013) at the Jesionowa mire in southern Poland noted the co-occurrence between the high percentage of *Sordaria* type and high percentages of Poaceae, Cerealia, *Rumex acetosa*/*acetosella* type and *Plantago lanceolata*, i.e., taxa characteristic of open areas that we observed in our pollen dataset during and after the outbreak. However, in the surroundings of the Jesionowa mire, the landscape has not opened up due to deforestation, but the grazing of livestock has intensified. Synchronously, Barabach (2014) reported a massive emergence of Glomeromycota spores, which can be widely considered an indicator of soil erosion (Ejarque et al., 2010; Van Geel et al., 1989). Indeed, the deforestation associated with the outbreak resulted in increased water and wind erosion. However, Kołaczek et al. (2013) argue that Glomeromycota spores can be considered indicators of soil erosion only in lacustrine deposits. In peatlands, there is a high risk of the presence of plant species capable of forming arbuscular mycorrhizae. Glomeromycota spores then come from fungi that have colonized the roots of plants growing on the surface of the peatland.

In their study of the Rzecin peatland, Milecka et al. (2017) reported an increase in charcoal in ca. 1910-1925 cal. CE. The authors linked this increase to the fires occurring in the Noteć Forest in the 1920s and 1930s. Still, it could also result from cleanup activities after the *P. flammea* outbreak, such as raking and burning litter with dead caterpillars. Barabach (2014) reported a higher content of ash and a higher charcoal concentration in the concerned interval. We did not observe increased micro- or macroscopic charcoal concentrations in the Miały peatland. It is possible that the redistribution of charcoal particles to the edges of the peatland occurred due to high water levels. A core taken closer to the edge could, therefore, give a complete answer as to the extent of burning.



Following the outbreak, an increase in the proportion of *Picea abies* until the early 1970s is
observed in our dataset. After the outbreak, initial management plans included diversification
of species composition in the newly planted forest's forest stands. Still, *P. sylvestris* was selected
as the primary species again. Other planted species included *Betula* (mainly along the roads),
*Pinus strobus*, *Pinus banksiana*, *Pinus rigida*, *Alnus glutinosa*, *Robinia pseudoacacia*, and
*Prunus serotina* (Mroczkiewicz, 1933). Considering that *P. abies* reaches sexual maturity after
20-30 years in open areas (Skrøppa, 2003) or even later in closed areas (~40 years) (Matthias
and Giesecke, 2014; Rispens, 2003), we conclude that the observed increase in *P. abies* pollen
is an echo of the 1922-1924 outbreak.
Recognizing the ecology of past *Panolis flammea* outbreaks in Central and Eastern Europe can
help model and predict its risk of occurrence in Northern Europe, which is warming due to
climate change. Pulgarin Díaz et al. (2022) (Pulgarín Díaz et al., 2022) report that between 1970
and 2020, the range of *Panolis flammea* in Finland shifted nearly 5° northward, 50 years earlier
than assumed. The remains of these butterflies could help determine the scale and ecology of
historical outbreaks in Central and Eastern Europe and thus better predict their future effects in
Northern Europe. Unfortunately, they do not preserve well in the sediment (Bąk et al., 2024).
We also haven't encountered them at Miały peatland. Palaeoecological analyses such as pollen
and testate amoeba analyses can support recognising the results of such historical outbreaks,
but they do not provide an answer that an outbreak occurred. There are, however,
palaeoecological reconstructions of outbreaks caused by other pests whose remains are better
preserved in the sediment. Schafstall et al. (2022) showed the usefulness of subfossil bark
beetles for reconstructing disturbances occurring in *Picea abies* forests in Slovakia.

**Changing trophic conditions as an effect of post-outbreak forest management**
The effect of the *Panolis flammea* outbreak was tens of thousands of hectares of damaged
forests. Damaged forests were cleaned, and the land was prepared for new plantings. However,
the opportunity to rebuild the forest's species structure was not seized. Easy-to-manage and fast-
growing pine trees were used for forest regeneration (Ankudo-Jankowska, 2003), which caused
a change in the trophic conditions of peatland. After the infestation, in our dataset, we primarily
notice the expansion of *Sphagnum* mosses, which displace monocotyledonous plants.
*Sphagnum* content reaches 65% for ca. 1900-1925 cal. CE and already 85% for ca. 1955-1960
cal. CE, further increasing in the upper part of the section (Fig. 3). The development of
*Sphagnum* mosses was possible by more acidic conditions. *Sphagnum* mosses, which adapted
to acidic conditions, won the competition with other plants. We also note the decrease of pH in



our data (Fig. 3). We assume that more acidic conditions in the peatland after *Panolis flammea*
outbreak are the result of monoculture plantings after this devastating event. Many studies
document the ability of various pine species to acidify the soil (Berthrong et al., 2009;
Cifuentes-Croquevielle et al., 2020; Hornung, 1985; Turner and Lambert, 1988). Our
assumption is confirmed by the period of occurrence of the maximum of the *Pinus sylvestris*
pollen curve at Miały, which is in the 1950s and 1960s. This is because *Pinus sylvestris* in dense
forest complexes begin flowering at the age of about 25-30 years (Mátyás et al., 2004).
The change in trophic conditions at this time is also documented by the completely disappearing
Cyanobacteria and algae (Fig. 4), which indicates that the peatland was cut off from the
groundwater supply. This observation is supported by the concurrent change in the Nd isotopic
signatures towards higher values (Fig. 2).
In the period of the transition of trophic conditions in a peatland, we observed the appearance
of *Bryophytomyces sphagni* (HdV-27). Some studies point out that this fungus is an indicator
of the change from minerotrophic to ombrotrophic conditions in a peatland, especially in
association with the appearance of *Sphagnum* spores (van Geel et al., 2020). Although we
observe numerous spores of this fungus in the narrow period of changing trophic conditions in
our dataset, we also note that the massive number of *B. sphagni* spores does not necessarily
indicate sudden ombrotrophication of the peatland. There are many studies where the
appearance of *B. sphagni* does not correlate with the ombrotrophication of the peatland (van der
Linden et al., 2008; McCarroll et al., 2017; Yeloff et al., 2007). Thus, we emphasise the need
for better recognition of the ecology of *B. sphagni*. With the appearance of *B. sphagni*,
*Gaeumannomyces caricis* (HdV-126) disappear. *G. caricis* is a fungus associated with *Carex*
(van Geel and Aptroot, 2006; Pals et al., 1980). In our plant macrofossil data, *Sphagnum*
mosses, as we mentioned above, have almost completely displaced monocots, including *Carex*,
which dominated the peatland in the phases 5, 4, and 3. A coincident disappearance of *G.*
*caricis*, the appearance of *B. sphagni* and the development of *Sphagnum*, has been noted in the
past in southwest France (Aoustin et al., 2022).
Sudden changes in trophic conditions, resulting in subsequent changes in the vegetation cover
in the catchment, are one of the most common causes of critical transitions in peatlands
(Lamentowicz et al., 2019b).
**Fire in 1992 - the second-largest fire in the post-World War II history of Poland**



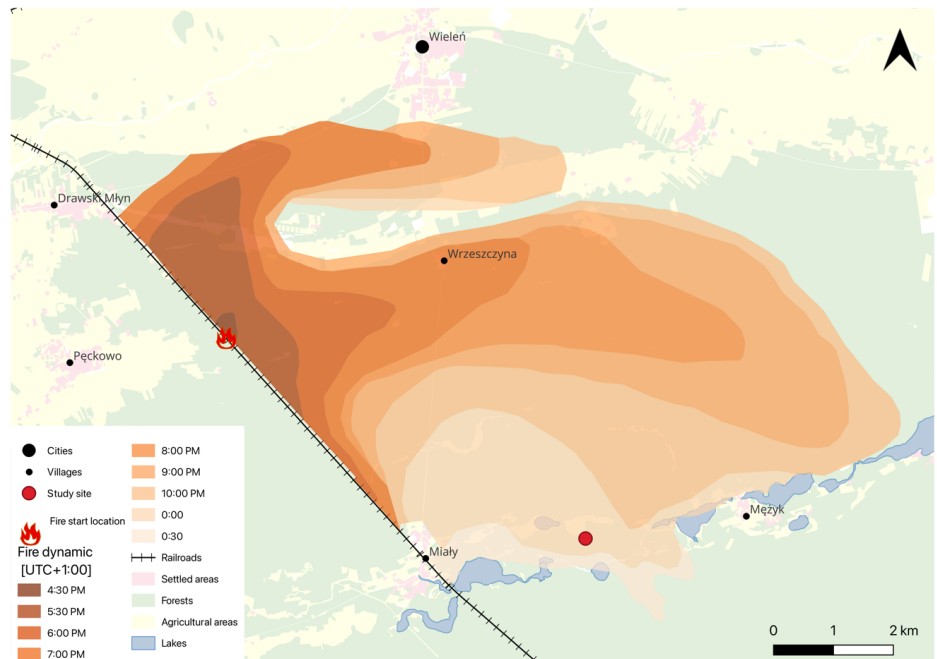

Fig. 6. The rate of fire spread in the Noteć Forest in 1992.

Potential high and medium fire danger concerns 83% of forests in Poland (65% in Europe) (Szczygieł, 2012). This is mainly due to poor habitats and a homogeneous forest structure, with *Pinus sylvestris* as the dominant species. *Pinus*, in turn, favours the accumulation of a significant amount of dry biomass on the surface. Fire danger is also a result of the young age of the tree stands, which have not yet developed stable ecosystem links. The young stands result from planned forest management involving rapid wood harvesting and 20[th]-century ecological disasters (particularly insect outbreaks). Industrial pollution, increasing accessibility to the public, and climate change, resulting in prolonged droughts and water deficits, amplify the problems of forest composition and management.

The 1992 droughts were marked by fires in many regions of Poland (Polna, 2005) and other countries in central Europe (Kula and Jankovská, 2013; Somsak et al., 2009). Almost 12,000 forest fires were recorded in Poland alone, and nearly 48,000 ha of forest area burned. The largest fire in Poland's post-war history occurred near the town of Kuźnia Raciborska (Silesia, southern Poland) from 26 to 30 August. More than 9,000 ha of forest were destroyed (Szczygieł, 2012). Two weeks earlier, the second largest fire in Poland's post-war history had affected the Noteć Forest.



In the 1970s, Hernik (1979) and Ratajszczak (1979) signalled that the tree stands of the Noteć
Forest were weakened by repeated insect outbreaks. The authors stressed the need to introduce
admixture species to change the age structure of the forest and reduce the threat. Their
predictions soon turned out to be very accurate. June 2, 1992, a fire covered about 700 hectares
of the Noteć Forest, 400 hectares of which burned completely (Bugaj, 1992), and on August 10,
the fire consumed more than 5,000 hectares of forest in just eight hours (Fabijański, 1996), and
the area affected was mapped in detail by the foresters (Fig. 6). Only an enclave of several
hectares of deciduous old-growth forest resisted the fire. This event roughly coincides with the
period of substantial rapid change identified by the PrC curve (Fig. 5), suggesting that this
change may have contributed to the rapid alteration of the forest ecosystem reflected in pollen
record.
Macroscopic charcoal concentrations did not register this fire event as we expected. Although
the concentration of microscopic charcoal in 1989-1991 cal. CE (ca. 30,800 particles/cm$^3$) and
1991-1994 cal. CE (ca. 27,500 particles/cm$^3$) is higher than in the 1986-1989 cal. CE (ca.
10,000 particles/cm$^3$) and 1994-1997 cal. CE (ca. 16,300 particles/cm$^3$), these values do not
reflect the actual scale of the forest destruction, especially since the fire took place near the
peatland (Fig. 5). A smaller-than-expected signal from the 1992 fire in charcoal analysis was
also obtained by Barabach (2014) in the nearby Rzecin peatland. The small amount of
macroscopic charcoal may be explained by the fact that the more intense the fire, the smaller
the charcoal particles it produces (Schaefer, 1973). Additionally, before the particles are
deposited, their dispersion by wind and water plays an important role (Patterson et al., 1987).
By the time the fire reached the peatland, heavy rain had fallen, reaching a value of 31.5 mm
(Institute of Meteorology and Water Management, 2025). This rain stopped the smoke from
spreading further away, however, it reached the Miały peatland (Fig. 6).
The events are, however, well recorded by other proxies. Directly after the fire – 1991-1994
cal. CE and 1994-1997 cal. CE – a substantial decrease in the percentage of arboreal pollen,
especially of Scots pine, is observed in the pollen dataset. At the same time, the *Pinus* stomata
appear, which may indicate a fall of needles to the surface. However, we recommend a cautious
approach to interpreting the presence of *Pinu*s stomata. While burnt *Pinus* stomata would give
certainty to the occurrence of fire, needle fall due to other processes should also be considered.
High water levels also may have contributed to the shedding of needles by *Pinus* in the peatland
(which we explain below). *Rumex acetosa/acetosella* type reaches its maximum percentage,
which is accompanied by an increase in the percentage of pollen of Poaceae, a taxon
characteristic of open areas, indicating the landscape's opening due to the forest's reduction.



The water table rose to the ground level, probably due to inundation. The rise in the groundwater
level shortly after increased fire activity is a well-known phenomenon observed at other sites
(Marcisz et al., 2015). The rise in water level is correlated to a high concentration (72%) of the
testate amoeba *Galeripora discoides*, which tolerates hydrologically unstable conditions and is
abundant in disturbed ecosystems (Lamentowicz and Mitchell, 2005). Therefore, we note that
it is not always possible to unambiguously identify local fire events from even high-resolution
charcoal analysis and that historical sources can validate the data. This is a crucial finding
regarding the interpretations of paleofire reconstructions, pointing out that even catastrophic
fires can go unnoticed in the sedimentary record.
The scale and frequency of catastrophic fires, including forest and peatland fires, have been
increasing worldwide for decades due to climate change (Sayedi et al., 2024). In terms of the
total area burnt, the year 2022 was the second-worst year ever recorded in the European Union
(San-Miguel-Ayanz et al., 2023). Nearly 900,000 ha of natural areas were burned, 43% of which
were located in Natura 2000 sites. In Poland, almost 7,000 fires of natural areas (including more
than 4,800 forest fires) were recorded resulting in approximately 2,850 ha of area burnt
(including 2,210 ha in forests). In terms of the number of fires in natural areas, more fires were
recorded only in France (22,800 fires; 70,300 ha), Spain (10,500; 268,000 ha), and Portugal
(10,400; 110,000 ha). Forest fires in Poland were, therefore, frequent but covered small areas
(0.4 ha/fire on average). Most fires in Poland occurred in May (more than 25%). This pattern is
vital when compared with dendroclimatic data. A recent study from the pine-dominated Tuchola
Forest in Poland revealed a negative correlation between Scots pine growth and rainfall in May
(Bąk et al., 2024). A water deficit in May carries, therefore, many dangerous consequences. In
2022, there were 84 fire incidents in the Noteć Forest resulted in 8.4 ha of burnt area. From
2007 to 2022, there were more than 1170 fire incidents covered 96.7 ha. Hence, the Noteć Forest
is a high-fire-risk area and, as a large monoculture forest complex, requires continuous
monitoring, including within EU structures.

### 763   Conclusions

Understanding the functioning of peatlands that are under severe climatic pressure and exposed
to extreme events in recent decades is crucial for their conservation and monitoring. Peatlands,
as archives of environmental change, are sources of valuable information about past ecological
disasters, recorded in both the palaeoecological and geochemical records. Combining these two
approaches gives a complete picture of environmental changes due to fires or insect outbreaks.



The conclusions of such studies can be successfully used to predict the consequences of contemporary phenomena. Particularly severe disasters can even lead to peatland ecosystems reaching critical transitions, after which there is an irreversible change in hydrological and trophic conditions, followed by a change in vegetation. We have identified many paleo-indicators that allow a comprehensive assessment of the peatland's response to catastrophic events both at the time of these events and on a long-term scale (Fig. 7).

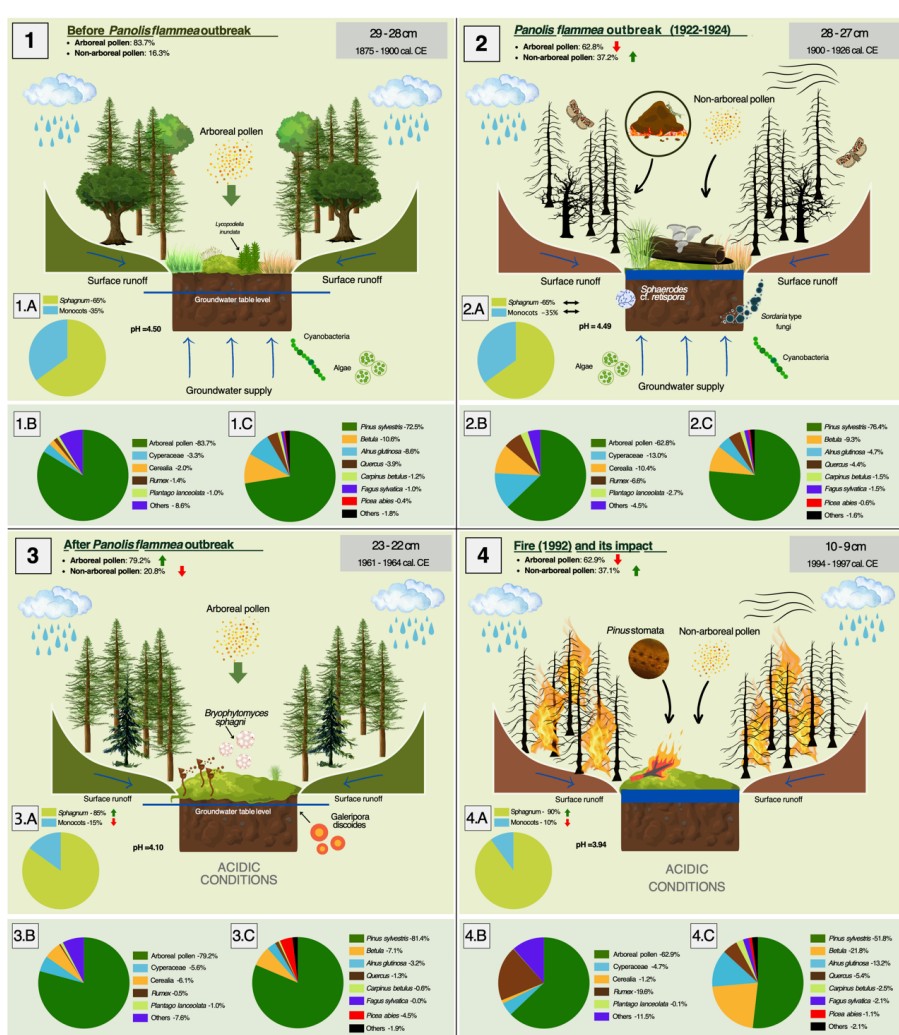

Fig. 7. Diagram showing environmental changes in the Miały peatland and the forest surrounding it as a result of the *Panolis flammea* outbreak (1922-1924; boards no 1 and 2),



leading to a change in forest structure to a *Pinus sylvestris* monoculture (3) and the
consequences of poorly resilient monocultures in the form of the 1992 fire (4). The percentages
of taxa in the pie charts were taken from palynological data. Each of the four boards corresponds
to one specific layer in the peat profile – the depth of the layer and the calibrated period are
marked in the upper right corners of the boards in the grey box.

We have shown that the peatland has rapidly acidified as a result of *Panolis flammea* infestation
and forest restoration activities. We reported a significant decrease in *Pinus sylvestris* pollen
during catastrophic events. Competition among plants in the peatland was won by those adapted
to acidic conditions *Sphagnum* mosses, which displaced monocotyledonous plants. We point
out that it is difficult to identify past *Panolis flammea* outbreaks, as the remains of these
butterflies do not preserve well in sediments. We emphasized a cautious approach to fungi as
bioindicators of environmental change due to many ambiguous interpretations in studies.
Charcoal analysis can provide information on localized fires, but it should be emphasized that
not every fire is recorded in this way. For this reason, adequate validation of the data with
historical sources or, if these do not exist, multi-proxy palaeoecological analyses are essential.
However, we point out that other paleo-recordings, treated cautiously, can help identify past
fires, such as *Pinus* stomata. To understand current or recent changes in peatlands and their
surroundings, it is often not enough to analyze the last hundred or two years, but the background
coming back hundreds or thousands of years must be considered. Only such a combination
gives a complete overview of changes due to human activity, climate change or ecological
disasters. We observed that there has been no catastrophic deforestation for more than 1,800
years. Major deforestation occurred only after changes in forest management. The peatland was
also hydrologically and trophically stable for most of the time analyzed. Drastic changes in
these conditions have occurred due to the *Panolis flammea* infestation and its consequences.

**Data availability**
The open dataset that supports the findings of this study is available in Mendeley Data with the
identifier doi: 10.17632/cv5t59wf24.1

**Appendix**
**Appendix Figure 1**



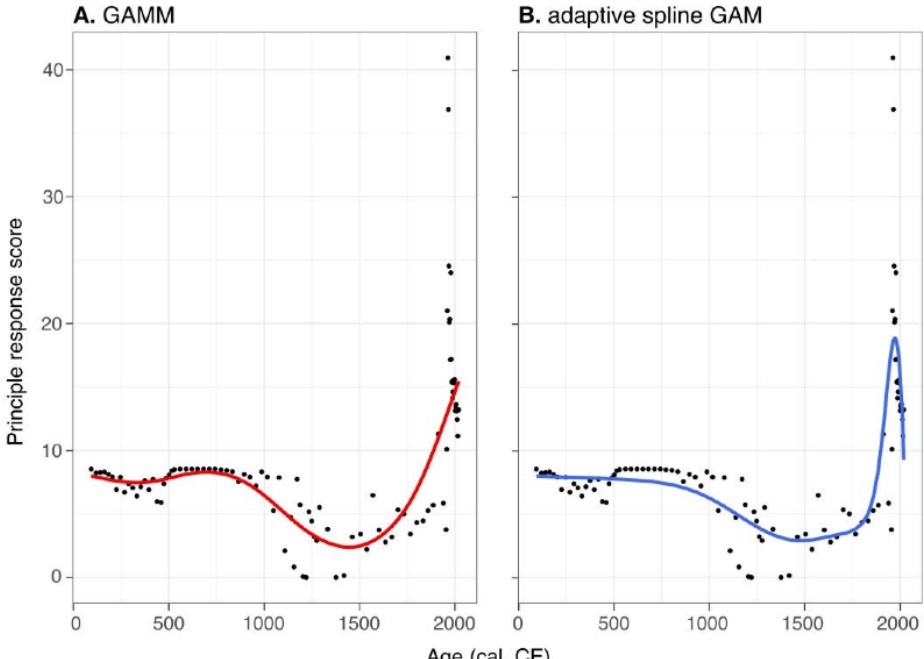


Figure A1. PrC (A) GAMM and (B) GAM with adaptive spline - raw scores and fitted
relationships.

## Authors contribution

MB – fieldwork, laboratory analyses (bulk density, carbon accumulation, plant macrofossils, selection of plant macrofossils for AMS radiocarbon dating), age-depth modelling, data interpretation, visualization, writing (original draft)

ML – fieldwork, support in plant macrofossil analysis, data interpretation, writing (commenting and editing)

PK – laboratory analyses (pollen and spores), age-depth modelling, data interpretation, visualization, writing (commenting and editing)

DW – laboratory analyses (testate amoebae), testate amoeba-based reconstructions, data interpretation

MJ – fieldwork, data interpretation, writing (commenting and editing)

LA – statistical analyses, data interpretation, writing (commenting and editing)

KM – funding acquisition, conceptualization, fieldwork, laboratory analyses (charcoal), data interpretation, visualization, writing (commenting and editing)



**Competing interests**

The authors declare no competing interests.

**Acknowledgements**

The study was funded by the National Science Centre, Poland, grant 2020/39/D/ST10/00641.
We thank Jay Tipton for his help in the field and Małgorzata Suchorska for the laboratory
preparation of pollen samples.

**Financial support**

This research has been supported by the Narodowe Centrum Nauki (grant no.
2020/39/D/ST10/00641).

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
