# Peer review of "20th-century ecological disasters in central European monoculture pine plantations led to"

_EGUsphere, 2025_

## Author Response (AR1)

**Responses to review No. 1**

**Reviewer general comment:** *This multi-proxy palaeoecological study assesses how one of Poland's largest peatlands, currently surrounded by monoculture Scots pine forests, was impacted by two major environmental disasters - the 1922-1924 Panolis flammea outbreak, and the historic 1992 fire. The authors have compelling evidence to support the environmental impacts of these major disasters and their implications regarding modern forest management. My only concerns with the paper are 1) the font in all the figures are too small and thus really hard to read; and 2) the current discussion section is a bit confusing and needs to be restructured. I hope my specific comments below will help with both of those concerns.*

**Author's response**: We thank to the reviewer for his/her comments. Undoubtedly, they were all helpful in improving the quality of the manuscript, organizing its structure and linguistic correctness. Below are our responses to each of them and our actions to improve the article.

**Line 42: simplified linkages to what? Are you trying to say simplified diversity here?**

**Author's response**: Indeed, the sentence was constructed imprecisely and was unclear. It is about simplified linkages in the food web, which result in the ecosystem being less resilient to various types of disturbances.

**Actions**: We corrected the sentence's meaning, clarifying what linkages we meant. We hope that after the correction, this part of the text is no longer in doubt.

**Before**: *The danger is even higher for peatlands located within monoculture tree plantations that have simplified linkages […] and thus are more sensitive to fires, strong winds, droughts, and insect outbreaks.*

**After**: *Such an environment is particularly dangerous for Poland's peatlands because monoculture tree plantations have simplified linkages in food webs and thus are more sensitive to fires, strong winds, droughts, and insect outbreaks (Chapin et al., 2012), which also causes a threat to peatlands.*

**Lines 42-43. Neither the Seidl et al. citation nor the Westerling citation discuss monoculture tree plantations or peatlands. I respectively ask the authors to either rewrite the sentence so it appropriately captures the scientific findings of these citations, or find more appropriate references that support the statement, " monoculture tree plantations... are more sensitive to fires, strong winds, droughts, and insect outbreaks."**

**Author's response**: The citations Seidl et al, 2014 and Westerling, 2016 refer to the increase in the frequency of extreme events, not to the sensitivity of pine monocultures to extreme events.

**Actions**: We have edited and corrected the sentences to leave no doubt as to which citations refer to which information. The correction resulted in two separate sentences to which the corresponding citations were assigned.

**Before**: *The danger is even higher for peatlands located within monoculture tree plantations that have simplified linkages (Chapin et al., 2012) and thus are more sensitive to fires, strong*

*winds, droughts, and insect outbreaks that are more common in recent years (Seidl et al., 2014; Westerling, 2016).*

*__After__: Such an environment is particularly dangerous for Poland's peatlands because monoculture tree plantations have simplified linkages in food webs and thus are more sensitive to fires, strong winds, droughts, and insect outbreaks (Chapin et al., 2012), which also causes a threat to peatlands. It should be strongly emphasized here that such extreme phenomena have become more common in recent years around the world (Seidl et al., 2014; Westerling, 2016).*

**Line 258: Should be "generalised additive mixed models."**

**Author's response**: Indeed, the abbreviation expansion was wrong.

**Actions**: An error in the GAMM abbreviation expansion has been corrected from mixed model generalised additive models to generalised additive mixed models

**Before**: *Mixed model generalised additive models (GAMMs) were then fitted to the data, with a smoothing term accounting for temporal autocorrelation.*

**After**: *Generalised additive mixed models (GAMMs) were then fitted to the data, with a smoothing term accounting for temporal autocorrelation.*

**Line 259: Can you be more specific with your "smoothing term?" Did you use a specific family and link function, or apply a k-function? Did you use REML?**

**Author's response**: We appreciate the reviewer's request for clarity in the methodology used in our rate of change analysis. We opted to use the same methodology used in Burge *et* al. (2023) - a cubic regression spline was used as the smoothing basis, with $k = 20$. We tested a range of values for $k$ to ensure the model avoids overfitting or underfitting the data. Likewise, we did not use REML, using instead Maximum Likelihood (ML) for consistency with Burge's framework. I just reanalysed the data using REML in place of ML as a smoothing parameter, and it didn't make an appreciable difference to the results.

**Actions**: A couple of lines containing the above information have been added to the manuscript (lines 260-265) for additional clarity and transparency, reducing the need to refer to or be familiar with the original framework by Burge *et* al. (2023).

**Before**: *Mixed model generalised additive models (GAMMs) were then fitted to the data, with a smoothing term accounting for temporal autocorrelation. [...]*

**After**: *Generalised additive mixed models (GAMMs) were then fitted to the data, with a smoothing term accounting for temporal autocorrelation. A cubic regression spline was used as the smoothing basis, with k = 20. A range of values for k was tested to ensure the model avoids overfitting or underfitting the data. Likewise, Maximum Likelihood (ML) was used for consistency with Burge's framework, instead of REML (Restricted Maximum Likelihood). However, REML was used to reanalyse the data in place of ML as a smoothing parameter, and it didn't make an appreciable difference to the results. [...]*

**Figures 2-4: the font size is currently too small to see. Can you increase font size. You might have to eliminate some data you don't discuss in the figure to accommodate a larger font size.**

**Author's response**: We agree with the reviewer's opinion.

**Actions:** We have corrected the figure by making the font larger. We have made the figure clearer. We also reduced the number of data presented by eliminating some of the dates from the timeline, as well as some of the values from the horizontal axes next to the curves.

In Figure 2, we made the font larger, and we removed part of the values from the horizontal axes and the age axis.

In Figure 3, we made the font larger, we removed non-discussed taxa curves (*Cirsium* – fruits), and some of the values from the horizontal and vertical axes.

In Figure 4, we made the font larger, we removed non-discussed taxa curves (*Salix*, *Populus*, *Tilia cordata*, *Abies alba*, *Acer*, *Pteridium aquilinum, Melampyrum*, *Urtica*, *Humulus/Cannabis*, *Centaurea cyanus*, *Fagopyrum esculentum* type, *Ambrosia artemisiifolia* type, Chenopodiaceae, *Plantago major*, Apiaceae undiff, *Potentilla* type, Brassicaceae, *Anthemis* type, *Aster* type, Cichoriaceae, *Galium* type, *Typha latifolia*, *Sparganium* type, *Drosera rotundifolia*, HdV-153 *Riccia*, Filicales monolete, *Scenedesmus*, *Spirogyra* type, *Mougeotia* type. *Pediastrum* – sum, HdV-128A, HdV-128B, HdV-1 *Gelasinospora* sp., HdV-30 *Helicoon pluriseptatum*, HdV-28 Copecoda spermatophores, HdV-31 *Archerella flavum*, HdV-32A *Assulina muscorum*, HdV-32B *Assulina seminulum*), and some of the values from the horizontal and vertical axes.

**Figure 4: What is the significance of the *Betula alba* type pollen curve? I think an admixture or deciduous curve would be more appropriate so the reader can visually see the decline in these pollen type during the specific discussion points mentioned in the discussion.**

**Lines 478-481: An admixture or deciduous curve in Figure 4 would be extremely helpful to better see these points.**

**Author's response**: Thanks to the reviewer for the comments. Indeed, the deciduous tree curve would be very helpful in understanding the changes in the composition of the forest. We agree that highlighting the *Betula alba* type curve was not necessary.

**Actions**: We modified the diagram (Figure 4). We created a deciduous tree curve as the reviewer suggested (on the left side). We moved the *Betula alba* type curve to a set of other curves relating to trees and shrubs. The curve of deciduous trees indicates a decline in the share of these taxa in the composition of the forest.

**Modified figures we presented below.**

**Figure 2**

**Before**:

[Figure]

**After**:

[Figure]

**Figure 3**

**Before:**

[Figure]

**After:**

[Figure]

**Figure 4**

**Before:**

[Figure]

**After:**

[Figure]

**Line 310:** *Carpinus betulus* **is missing a percent sign.**

**Author's response**: We thank the reviewer for pointing out this missing element.

**Actions**: We have completed and corrected the percentage value.

**Before**: *Pinus sylvestris (39.0–65.8%) grains are the most frequent, but the pollen of deciduous trees is relatively common as well (Fig. 4): Betula (7.4–26.4%), Alnus (max. 17.0%), Quercus (max. 15.6%), Carpinus betulus (max. 5.8), Corylus avellana (max. 4.6%), [...]*

**After**: *Pinus sylvestris (39.0–65.8%) grains are the most frequent, but the pollen of deciduous trees is relatively common as well (Fig. 4): Betula (7.4–26.4%), Alnus (max. 17.0%), Quercus (max. 15.6%), Carpinus betulus (max. 5.8%), Corylus avellana (max. 4.6%), [...]*

**Lines 452-455: I apologize, I'm not familiar with PrC. How do you know which line belongs to the PrC, the GAMM and the adaptive spline in Figure 5? Can you provide more detail here so Figure 5 is easier to understand.**

**Author's response**: We appreciate the request from the reviewer for clarity in Figure 5 and its caption, now realising that the caption does not contain adequate information to allow the figure to stand alone apart from the article text.

**Actions**: In Figure 5, the raw PrC scores are displayed as points. The line (both black and red sections) represents the GAM fit which is overlain across the raw points. The red section of the line indicates where the confidence intervals surrounding the first derivative of the GAM did not include zero – representing a period of significant, rapid rate of change in the pollen data. We've now updated the figure caption to reflect the information detailed above and included a legend to Figure 5 for added clarity- this information was included but was not well explained. We have also added some information to the methodology section to line #, further explaining the methods used, and better representing the package by Burge et al., and its dependencies.

**Before:** *Periods of significant change were identified in the GAMM models by calculating the time intervals where the confidence intervals surrounding the first derivative did not include zero.*

**After:** *Periods of significant change were identified in the GAMM models by calculating the time intervals where the confidence intervals surrounding the first derivative did not include zero. PrC curves were derived from constrained ordination of the time series palynological data, which use the prcurve() function (package analogue) in R.*

The modified figure is presented below.

**Figure 5**
**Before***:*

[Figure]

**After**:

[Figure]

**Discussion section 1: In its current form, this discussion section seems to mix objective 1 with 2. I had a hard time following this section specifically because you don't fully discuss things in depth; because they are in the following discussion section. I think this discussion section needs to be re-written; I think lines 468-501 are fine for this discussion section, but you should really end with which events are unprecedented in your record. That would then lead into your following discussion sections where you discuss in detail each of the two major events.**

**Author's response:** We thank the reviewer for this comment. We agree that the discussion was not clear, and some of the information was intertwined, duplicated, or referred to other sections of the discussion.

**Action:** The discussion has been rewritten according to the reviewer's suggestions. The information that relates to acidification has been moved to the section on changing trophic and hydrological conditions after *Panolis flammea* infestation. We have provided detailed explanations of the changes in the responses to individual comments below. We hope that these will be satisfactory to the reviewer.

**Line 496: Sentence suggestion: "...the distinct admixture of *Quercus*, *Carpinus betulus*, and *Corylus avellana* was recognized in our study." Also, just curious, how do you know these weren't wind-drifted from regional sources?**

**Author's response**: We thank the reviewer for suggesting editorial changes, being curious and willing to explore the topic. Indeed, the palynological method has the feature of recording both regional and local signals. However, this means that throughout the core, the proportion of local and regional sources should be about the same. In addition, in a closed landscape (as in this case - a forested landscape), the proportion of pollen supplied from close distances is relatively high. We therefore assume that each sample of our core represents a high share of pollen from close distances.
The size of the peatland is also important. The proportion of pollen from a local source to pollen from a regional source in a small peatland is higher than in a large peatland. Our peatland is only 1.4 hectares in size.

**Action**: We have corrected the sentence as suggested by the reviewer.

**Before:** *[…] the distinct admixture of Quercus, Carpinus betulus, and Corylus avellana was recognized.*

**After:** *[…] the distinct admixture of Quercus, Carpinus betulus, and Corylus avellana was recognized in our study.*

**Lines 496-497: Sentence suggestion: "Previous multi-proxy palaeoecological studies exist from the Noteć Forest, however, those previous were unable to..."**

**Author's response**: We appreciate the reviewer's work on the grammatical and stylistic correctness of the manuscript. We agree that the proposed sentence form is clearer.

**Action**: We have corrected the sentence as suggested by the reviewer.

**Before:** *The other multi-proxy palaeoecological studies from the Noteć Forest were unable to provide such information because […]*

**After:** *Previous multi-proxy palaeoecological studies exist from the Noteć Forest; however, those previous ones were unable to provide such information because […]*

**Lines 501-502: Sentence suggestion: No need for a new paragraph here since there is no new topic.**

**Author's response**: We appreciate the reviewer's work on the grammatical and stylistic correctness of the manuscript and its clarity. We agree that creating a new paragraph was unnecessary.

**Action**: We have combined the two paragraphs into one, as they cover the same topic.

**Before:** *[…] and did not capture the entire background of the changes related to human activity and subsequent forest management (Barabach, 2014; Lamentowicz et al., 2015; Milecka et al., 2017; Słowiński et al., 2019).*
*The knowledge of the historical background is essential for the interpretation of the ecosystem response to forestry practices because […]*

**After:** *[…] and did not capture the entire background of the changes related to human activity and subsequent forest management (Barabach, 2014; Lamentowicz et al., 2015; Milecka et al., 2017; Słowiński et al., 2019). The knowledge of the historical background is essential for the interpretation of the ecosystem response to forestry practices because […]*

**Line 511: delete 'the' in this sentence; "All these taxa disappeared in phase 5.'**

**Author's response**: We appreciate the reviewer's work on the grammatical correctness of the manuscript. We agree that the proposed sentence form is proper.

**Action**: We have removed the 'the' from this sentence.

**Before:** *All these taxa disappeared in the phase 5, after ca. 1960 CE.*

**After:** *All these taxa disappeared in phase 5, after ca. 1960 CE.*

**Line 513: delete 'the' before *Sphagnum* mosses; and 'acid' should be 'acidic'**

**Author's response**: We appreciate the reviewer's work on the grammatical correctness of the manuscript. We agree that the proposed changes are proper.

**Action**: We have removed the 'the' from this sentence, and we have corrected a wrongly spelt adjective.

**Before:** *Monocot plants and brown mosses were displaced by the expansion of the Sphagnum mosses that tolerate acid conditions.*

**After:** *After the infestation, in our dataset, we also notice the expansion of Sphagnum mosses, which tolerate more acidic conditions.*

**Lines 514-516: Does this mean acidification? I assume yes based on the opening of the next sentence, but it would be nice to synthesize the 'so what' of this data. Also, any mention of acidification (i.e., lines 512-524) could be moved to the next discussion section where you again discuss acidification of the landscape post the *Panolis flammea* outbreak; it would simplify and streamline your discussion instead of having to discuss the acidification process twice.**

**Author's response**: We thank the reviewer for his comments on the discussion. We agree that the solution of moving all content about acidification of the peatland and changing its trophic and hydrological conditions into one section is appropriate. We see that this will make the discussion clearer and more structured. We have decided to respond to the three comments together, as they were all about rewording the discussion and making it more structured.

**Action**: We have rewritten the discussion, moving some content on acidification and changing trophic and hydrological conditions into one section. In the first section, we only refer to the issue of ecosystem stability before the introduction of planned forest management. Then, we introduce information on extreme phenomena and their consequences in the results of this way of forest management. We synthesise, therefore, our data indicate acidification.

**Lines 519-520: "...to poor fen by combining *Sphagnum* sub. *Subsecunda* and *Lycopodiella inundata* taxa in phase 4 (ca. 1660-1960 CE)." Why did you combine those taxa? Is 'combining' a typo?Additionally, when I first read the paper, I immediately wanted to know why 1960 CE was important management wise. But then realised you discuss in detail the outbreak and its eventual contribution to acidification. I think having all discussion regarding the outbreak and acidification in one section would be much easier to understand.**

**Author's response**: Thank you to the reviewer for your comments on the structuring and clarity of the discussion. We appreciate the comments and have decided to make changes as necessary.

**Action**: We have removed sentences relating to the indicator significance of the taxa *Sphagnum* sub. *Subsecunda* and *Lycopodiella inundata* in determining transitions between peatland trophic conditions.

After rewriting the discussion, we have emphasised the importance and reasons for choosing 1960 as the date of the extreme change in trophic and hydrological conditions in the peatland. We hope that the discussion is now clear and structured.

As we mentioned above, some of the information from Section 1, according to the reviewers' comments, has been moved into one section where we discuss the causes and consequences of acidification of the peatland and rewritten.

**Line 529: 'pre-infestation part', this is the first mention of the 'outbreak.' Without any context, this sentence makes no sense. Thus, further evidence of why you should combine this paragraph with the next discussion section.**

**Author's response**: Thank you to the reviewer for your comments on the structuring and clarity of the discussion. We appreciate the comments and have decided to make changes as necessary.

As we mentioned above, some of the information from Section 1, according to the reviewers' comments, has been moved into one section where we discuss the causes and consequences of acidification of the peatland and rewritten.

**Line 558: "Over 500,000 ha of forest have been defoliated in Europe" as a result of this particular outbreak? Or is 500,000 ha total ha affected over the past decade? Be more specific with time here.**

**Author's response**: We appreciate the reviewer's work on the clarity of the manuscript. We agree that the sentence should be written more specifically.

**Action**: We have corrected a sentence by detailing the time scale as a reviewer suggested.

**Before:** *Over 500,000 hectares of forests have been defoliated in Europe (Głowacka, 2009).*

**After:** *As a result of the 1922-1924 Panolis flammea infestation, over* *500,000 hectares of forests have been defoliated in Europe (Głowacka, 2009).*

**Line 560: sentence suggestion: "Over the next two years, between 1922-1923, ca. 64,000 ha of the forest..."**

**Author's response**: We appreciate the reviewer's work on the grammatical and stylistic correctness of the manuscript. We agree that the proposed sentence form is clearer.

**Action**: We have corrected the sentence as a reviewer suggested

**Before:** *Still, in the following two years,* *ca. 64,000 hectares of the forest were destroyed (Hernik, 1979).*

**After:** _Over the next two years, between 1922-1924,_ ca. 64,000 hectares of the forest were destroyed (Hernik, 1979).

**Lines 563-566: sentence suggestion: "This outbreak is evidenced in our pollen record, marked a sharp decrease in the percentage of _Pinus sylvestris_ pollen (48%; 1900-1926 cal. CE) compared to the neighbouring layers - ca. 1875-1900 cal. CE (60.6%) and ca. 1925-1950 cal. CE (62.8%).**

**Author's response**: We appreciate the reviewer's work on the grammatical and stylistic correctness of the manuscript. We agree that the proposed sentence form is clearer.

**Action**: We have corrected the sentence as a reviewer suggested.

**Before:** _We assume that in the pollen record, this outbreak is well recognizable (1900-1926 cal. CE; phase 4). It is marked by_ a sharp decrease in the percentage of Pinus sylvestris pollen (48.0%) compared to the neighbouring layers – ca. 1875-1900 cal. CE (60.6%) and ca. 1925-1950 cal. CE (62.8%).

**After:** _This outbreak is evidenced in our pollen record, marked_ a sharp decrease in the percentage of Pinus sylvestris pollen (48.0%) compared to the neighbouring layers – ca. 1875-1900 cal. CE (60.6%) and ca. 1925-1950 cal. CE (62.8%).

**Line 567: Przebieg..., 1929 citation appears to be missing the rest of the citation?**

**Author's response**: The author of this reference is unknown. The first word in the citation is the first word of the title of the paper. In the bibliographic list, it appeared as „Anon: Przebieg i bilans katastrofy sówkowej w Wielkopolsce, Rynek Drzewny i Budowlany, 116, 3–4, 1929."

**Action:** We edited the item in the bibliographical list so that it begins with the first word of the title, according to the citation called out in the text.

**Before:** _Anon:_ Przebieg i bilans katastrofy sówkowej w Wielkopolsce, Rynek Drzewny i Budowlany, 116, 3–4, 1929.

**After:** Przebieg i bilans katastrofy sówkowej w Wielkopolsce, Rynek Drzewny i Budowlany, 116, 3–4, 1929.

**Line 621: sentence suggestion: delete 'again' at the end of the sentence.**

**Author's response**: We appreciate the reviewer's work on the grammatical and stylistic correctness of the manuscript. We agree that the proposed sentence form is proper.

**Action**: We have corrected the sentence as a reviewer suggested.

**Before:** _Still, P. sylvestris was selected as the primary species_ _again_.

**After:** _Still, P. sylvestris was selected as the primary species._

**Line 633: sentence suggestion: 'Unfortunately, they do not preserve well in sediments (Bąk et al., 2024)."**

**Author's response**: We appreciate the reviewer's work on the grammatical and stylistic correctness of the manuscript. We agree that the proposed sentence form is proper.

**Action**: We have corrected the sentence as a reviewer suggested.

**Before:** *Unfortunately, they do not preserve well in the sediment (Bąk et al., 2024).*

**After:** *Unfortunately, they do not preserve well in the sediments (Bąk et al., 2024).*

**Lines 647-649: suggest moving all mention of *Sphagnum* and acidification from the first discussion section here, somewhere. Also, delete 'already' before '85%".**

**Author's response**: Thanks to the reviewer for his comment. We have complied with the discussion comments earlier. We hope that now the discussion is clear and the information is arranged in a way that helps to understand the message of the discussion. We appreciate the reviewer's work on the grammatical and stylistic correctness of the manuscript. We agree that the proposed sentence form is proper.

**Action**: We have updated the name of the section, as some of the information from the first discussion section has been moved here. Here we discuss threads related not only to changing trophic conditions, but also to hydrological conditions.

We have corrected the sentence as a reviewer suggested.

**Before:** *Sphagnum content reaches 65% for ca. 1900-1925 cal. CE and already 85% for ca. 1955-1960 cal. CE, further increasing in the upper part of the section (Fig. 3).*

**After:** *Sphagnum content reaches 65% for ca. 1900-1925 cal. CE and 85% for ca. 1955-1960 cal. CE, further increasing in the upper part of the section (Fig. 3) [...]*

**Lines 655-656: sentence suggestion: This is confirmed by the highest percentages of *Pinus sylvestris* at Miały between 1950-1960.**

**Author's response**: We appreciate the reviewer's work on the grammatical and stylistic correctness of the manuscript. We agree that the proposed sentence form is proper.

**Action**: We have corrected the sentence as a reviewer suggested.

**Before:** *Our assumption is confirmed by the period of occurrence of the maximum of the Pinus sylvestris pollen curve at Miały, which is in the 1950s and 1960s.*

**After:** *This is confirmed by the highest percentages of Pinus sylvestris at Miały between 1950-1960.*

**Lines 663-664: When does 'the period of transition to trophic conditions" actually occur?**

**Author's response**: Thanks to the reviewer for the comment. Indeed, the sentence should be more precise and specify what time we mean.

**Action**: We have corrected the sentence as a reviewer suggested.

**Before:** *In the period of the transition of trophic conditions in a peatland, we observed […]*

**After:** *In the period of the transition of trophic and hydrological conditions in a peatland (ca. 1925-1960 CE), we observed […]*

**Line 667: Again, when does 'the narrow period of changing trophic conditions" occur?**

**Author's response**: Thanks to the reviewer for the comment. Indeed, the sentence should be more precise and specify what time we mean.

**Action**: We have corrected the sentence as a reviewer suggested.

**Before:** *Although we observe numerous spores of this fungus in the narrow period of changing trophic conditions in our dataset, we also […]*

**After:** *Although we observe numerous spores of this fungus in the narrow period of changing trophic and hydrological conditions in our dataset (ca. 1925-1960 CE), we also […]*

**Line 676: sentence suggestion: delete 'the' before phases 5, 4, 3. Suggest changing '5,4,3' to '3-5'.**

**Author's response**: We thank the reviewer for his work on the stylistic and grammatical correctness of the manuscript. We agree that corrections are needed.

**Action**: We have corrected the sentence as a reviewer suggested.

**Before:** *In our plant macrofossil data, Sphagnum mosses, as we mentioned above, have almost completely displaced monocots, including Carex, which dominated the peatland in the phases 5, 4, and 3.*

**After:** *In our plant macrofossil data, Sphagnum mosses, as we mentioned above, have almost completely displaced monocots, including Carex, which dominated the peatland in phases 3-5.*

**Lines 676-678. Did these authors attribute the appearance of *B. sphagni* to changes in trophic conditions or ombrotrophication? If so, that would strengthen your argument.**

**Author's response**: The authors of this study do not explicitly write about ombrotrophication or acidification of the peatland they studied. However, they indicate that the large number of *B. sphagni* spores was one of the key data points based on which they decided to distinguish an additional developmental phase of the studied site, which they called "Sphagnum bog."

**Action**: We have expanded on the details of the conclusions of the Aoustin et al. (2022) article. We detailed that they emphasized the indicator role of *B. sphagni* as a taxon to help identify changes in the bog.

**Before:** *A coincident disappearance of G. caricis, the appearance of B. sphagni and the development of Sphagnum, has been noted in the past in southwest France (Aoustin et al., 2022).*

**After:** *A coincident disappearance of G. caricis, the appearance of B. sphagni and the development of Sphagnum, have been noted in the past in southwest France (Aoustin et al., 2022). These authors, among others, based on the large number of spores of B. sphagni, decided to separate the developmental phase of the object they studied, which they referred to as Sphagnum bog (Aoustin et al., 2022).*

**Lines 686-687: Is the 'potential high and medium fire danger' specific to modern fires? Late Holocene fire? What fires in time?**

**Author's response**: We appreciate the reviewer's work on the clarity of the manuscript. We agree that the sentence should be written more specifically.

**Action**: We have corrected a sentence by detailing the time scale as a reviewer suggested.

**Before:** *Potential high and medium fire danger concerns 83% of forests in Poland (65% in Europe) (Szczygieł, 2012).*

**After:** *Potential high and medium modern fire danger concerns 83% of forests in Poland (65% in Europe) (Szczygieł, 2012).*

**Line 690: What do you mean by 'ecosystem links'? Links to what exactly?**

**Author's response**: Thank you to the reviewer for the comment. Indeed, the sentence was constructed imprecisely and was unclear. It is about simplified linkages in the food web, which result in less resilience of the ecosystem to various types of disturbances.

**Actions**: We corrected the sentence's meaning, clarifying what linkages we meant. We hope that after the correction, this part of the text is no longer in doubt.

**Before**: *Fire danger is also a result of the young age of the tree stands, which have not yet developed stable ecosystem links.*

**After**: *Fire danger is also a result of the young age of the tree stands, which have not yet developed stable ecosystem links in food webs.*

**Lines 697-701: sentence suggestion: The largest fires in Poland's post-war history, which burned more than 9,000 ha of forest (Szcyzgiel, 2012), occurred near the town of Ruznia Raciborksa (Silesia, southern Poland) between 26 to 30 August, 1992. Two weeks prior to this event, the second largest fire in Poland's post-war history affected Noteć Forest."**

**Author's response**: We thank the reviewer for his work on the stylistic and grammatical correctness of the manuscript. We agree that corrections are needed.

**Action**: We have corrected the sentence as a reviewer suggested.

**Before**: *The largest fire in Poland's post-war history occurred near the town of Kuźnia Raciborska (Silesia, southern Poland) from 26 to 30 August. More than 9,000 ha of forest were destroyed. Two weeks earlier, the second largest fire in Poland's post-war history had affected the Noteć Forest.*

**After:** *The largest fire in Poland's post-war history, which burned more than 9,000 ha of forest (Szczygieł, 2012), occurred near the town of Kuźnia Raciborska (Silesia, southern Poland) between 26 and 30 August. Two weeks prior to this event, the second largest fire in Poland's post-war history had affected the Noteć Forest.*

**Line 704: "...and reduce the threat..." threat to what exactly?**

**Author's response**: Thank you to the reviewer for the comment. Indeed, the sentence was constructed imprecisely and was unclear. It is about the fire threat.

**Actions**: We corrected the sentence's meaning, clarifying what threat we meant. We hope that after the correction, this part of the text is no longer in doubt.

**Before**: *The authors stressed the need to introduce admixture species to change the age structure of the forest and reduce the threat.*

**After**: *The authors stressed the need to introduce admixture species to change the age structure of the forest and reduce the fire threat.*

**Lines 705-706: How does a fire cover 700 ha but only burn 400 ha? I don't understand.**

**Author's response**: Thank you to the reviewer for the comment. Indeed, the sentence was constructed imprecisely and was unclear.

**Actions**: We have corrected the sentence according to the cited reference.

**Before**: *June 2, 1992, a fire covered about 700 hectares of the Noteć Forest, 400 hectares of which burned completely (Bugaj, 1992) […]*

**After**: *June 2, 1992, a fire covered about 700 hectares of the Noteć Forest (Bugaj, 1992) […]*

**Lines 707-708: sentence suggestion: ...(Fabijański, 1996). The total area affected was mapped in detail by foresters (Fig. 6)."**

**Author's response**: We thank the reviewer for his work on the stylistic and grammatical correctness of the manuscript. We agree that corrections are needed.

**Action**: We have corrected the sentence as a reviewer suggested.

**Before**: *[…] hours (Fabijański, 1996), and the area affected was mapped in detail by the foresters (Fig. 6).*

**After:** *hours (Fabijański, 1996). The total area affected was mapped in detail by the foresters (Fig. 6).*

**Lines 733-735: Move this sentence to after the high water level discussion.**

**Author's response**: We agree with the reviewer that moving this sentence after the discussion of water levels improved the message of the issue and made the paragraph more organized and clearer.

**Action**: We have moved the sentence as the reviewer suggested.

**Before**: *[…] by Pinus in the peatland (which we explain below). Rumex acetosa/acetosella type reaches its maximum percentage, which is accompanied by an increase in the percentage of pollen of Poaceae, a taxon characteristic of open areas, indicating the landscape's opening due to the forest's reduction. The water table rose to the ground level, probably due to inundation. The rise in the groundwater level […]*

**After**: *by Pinus in the peatland (which we explain below). The water table rose to the ground level, probably due to inundation. The rise in the groundwater level […]. Rumex acetosa/acetosella type reaches its maximum percentage, which is accompanied by an increase in the percentage of pollen of Poaceae, a taxon characteristic of open areas, indicating the landscape's opening due to the forest's reduction.*

**Line 785: "We have shown that the peatland has..." Which peatland? Remind the reader which peatland you are talking about.**

**Author's response**: Thank you to the reviewer for the comment. Indeed, the sentence was constructed imprecisely and should be more specific.

**Actions**: We have added the name of the peatland in the proper place in the sentence.

**Before**: *We have shown that the peatland has rapidly acidified as a result of Panolis flammea infestation and forest restoration activities.*

**After**: *We have shown that the Miały peatland has rapidly acidified as a result of Panolis flammea infestation and forest restoration activities.*

**Line 798: change "coming" to "going"**

**Author's response**: We appreciate the reviewer's work on the grammatical and stylistic correctness of the manuscript. We agree that the proposed sentence form is proper.

**Action**: We have corrected the sentence as the reviewer suggested.

**Before***: To understand current or recent changes in peatlands and their surroundings, it is often not enough to analyze the last hundred or two years, but the background coming back hundreds or thousands of years must be considered.*

**After:** *To understand current or recent changes in peatlands and their surroundings, it is often not enough to analyze the last hundred or two years, but the background going back hundreds or thousands of years must be considered.*

**Lines 800-801: This is your main conclusion. Tell me which times in the past your record sees major changes; summarize figure 7 in the conclusions.**

**Author's response**: We appreciate the reviewer's request to expand the conclusions to include a time scale and a summary of Figure 7. We agree that this arrangement is more suitable.

**Action**: We have expanded the conclusions to include a time scale of events, including the two extreme events discussed earlier, summarised in Figure 7.

**Before***: The peatland was also hydrologically and trophically stable for most of the time analyzed. Drastic changes in these conditions have occurred due to the Panolis flammea infestation and its consequences.*

**After:** *The peatland was also hydrologically and trophically stable for most of the time analyzed. Drastic changes in trophic and hydrological conditions of the Miały peatland began after the introduction of planned forest management in the late 18th century, weakening the forests' resilience to environmental disasters. Particularly extreme changes occurred with the 1922-1924 Panolis flammea and the subsequent approach from forest restoration after 30-40 years. Keeping the forest structure homogeneous in turn led to a huge fire in 1992 (Fig. 7).*

**Response to Review No. 2**

**Reviewer's general comment:** This study investigates how different disturbance events affected the forest structure and a peat bog (including its hydrological and biochemical properties) over the past 2000 years, and aims to recognize and standardize patterns that could be applied to other sites. Even though the discussion deviates quite far into describing the full details of the insect outbreak and the forest fire, which are not really necessary to understand the magnitude of these events, the manuscript is overall decent. Figure 7 is inspiring and a nice conclusion of the manuscript.

I am a bit skeptical that no attempt was made to explain the strong fluctuations in the vegetation composition, according to the PcA, and I do not agree with the conclusion that the PcA shows that the rapid change already started after 1775 AD.

**Author's response**: We thank the reviewer for the review of the manuscript. The reviewer's comments, suggestions and opinion are valuable and provide valuable support in improving the quality of the manuscript. We hope that the responses to the following questions and comments will be comprehensive and well-reasoned, and that all the authors' efforts to improve the manuscript will be duly recognized and appreciated by the reviewer. Below are the answers to all the reviewer's questions and comments, especially those that cause the most concern and relate to the statistical analysis.

Abstract: I found the abstract not informative enough. Which palaeoecological proxies? How long was the record that you studied? The outbreak of what?

**Author's response**: Thank you to the reviewer for this comment. We agree that the abstract was not informative enough. We have taken steps to complete the missing information by the reviewer's suggestions.

**Action**: We completed the information about the age of the peat core, the name of the peatland, the type of insect which caused the outbreak, and the types of palaeoecological analysis used in this study as the reviewer suggested.

**Before**: *[…] Here, we examined how a peatland in one of Poland's largest pine plantation complexes responded to some of the largest environmental disasters observed in the 20$^{th}$ century across Central Europe – the 1922–1924 outbreak and the 1992 fire. As a disturbance proxy, we used a multi-proxy palaeoecological analysis supported by a neodymium isotope record. […]*

**After**: *[…] Here, we traced a 2000-year history of the Miały peatland located in one of Poland's largest pine plantation complexes and we examined how this peatland responded to some of the largest environmental disasters observed in the 20$^{th}$ century across Central Europe – the 1922–1924 Panolis flammea outbreak and the 1992 fire. As a disturbance proxy, we used a multi-proxy palaeoecological analysis (plant macrofossils, testate amoebae, pollen, non-pollen palynomorphs, micro- and macrocharcoal) supported by a neodymium isotope record. […]*

**Introduction**

Please check the language again, if all statements are in written language and not spoken language, and if no articles are missing (I think some are). I think it would be worth to

**consider to move some sentences around to give the introduction a better, logical structure from beginning to end.**

**Author's response:** We thank the reviewer for his suggestions and comments on improving the quality of the introduction. We have checked and rewritten the text of the introduction based on the reviewer's comments below. We have followed all the comments, and detailed responses to them are also presented below.

**Line 39: I would replace the word 'precious' with a less personal adjective.**

**Author's response**: We appreciate the reviewer's work on the linguistic correctness of the manuscript. We agree that the proposed change is proper.

**Action**: We have corrected the sentence as the reviewer suggested, replacing the word 'precious' with 'valuable'.

**Before**: *This is particularly important because peatlands are precious ecosystems accumulating a third of the world's soil carbon stocks (Parish et al., 2008) [...]*

**After**: *This is particularly important because peatlands are valuable ecosystems accumulating a third of the world's soil carbon stocks (Parish et al., 2008) [...]*

**Line 42: Maybe "simplified linkages" can be explained in the text so that the reader doesn't have to open the referred article; it is not clear to me what these simplified linkages are.**

**Author's response**: Thank you to the reviewer for the comment. Indeed, the sentence was constructed imprecisely and was unclear. It is about simplified linkages in the food web, which result in less resilience of the forest ecosystem to various types of disturbances.

**Actions**: We corrected the sentence, explaining what linkages we meant. We hope that after the correction, this part of the text is no longer in doubt and is clear for the reviewer. We meant simplified links in food webs.

**Before**: *The danger is even higher for peatlands located within monoculture tree plantations that have simplified linkages [...] and thus are more sensitive to fires, strong winds, droughts, and insect outbreaks.*

**After**: *Such an environment is particularly dangerous for Poland's peatlands because monoculture tree plantations have simplified linkages in food webs and thus are more sensitive to fires, strong winds, droughts, and insect outbreaks (Chapin et al., 2012), which also causes a threat to peatlands.*

**Line 45 – 51 break up the flow of the introduction, which I think should move from "negative impacts" to "recognize how these impacts work and used to work"; the description of the forest structure in Poland could maybe be moved.**

**Author's response**: Thanks to the reviewer for the comment. We agree that the structure of the introduction should be more structured, so we have made a few changes to the text following the reviewer's suggestions.

**Actions**: We have rewritten much of the introduction following the reviewer's suggestion. We have arranged the structure of the text so that it is more logical, starting with the determinants of negative effects to the recognition of those effects. We have moved the description of the structure of Poland's forests before the information relating to the negative consequences resulting from this structure.

**Before**: *The danger is even higher for peatlands located within monoculture tree plantations that have simplified linkages (Chapin et al., 2012) and thus are more sensitive to fires, strong winds, droughts, and insect outbreaks that are more common in recent years (Seidl et al., 2014; Westerling, 2016). These negative impacts have been recorded for various peatlands, including those in Central and Eastern Europe (Leonardos et al., 2024; Łuców et al., 2021). Forests cover 31% of Poland's area, equivalent to 94,770 km$^2$ (Statistical Office in Białystok, 2023). More than half of this forest cover comprises coniferous forests dominated by Scots pine (Pinus sylvestris L.). It is mainly the result of planned forest management in modern-day Poland in the 19$^{th}$ and 20$^{th}$ centuries (Broda, 2000). Pine monocultures were easier to manage and grew faster on poor soils, securing the continuous supply of raw material for the growing timber industry (Broda, 2000).*

**After**: *Hundreds of thousands of hectares of peatlands in Poland are located in forests, as forests cover 31% of Poland's area, equivalent to 94,770 km$^2$ (Statistical Office in Białystok, 2023). More than half of this forest cover comprises coniferous forests dominated by Scots pine (Pinus sylvestris L.). It is mainly the result of planned forest management in modern-day Poland in the 19$^{th}$ and 20$^{th}$ centuries (Broda, 2000). Pine monocultures were easier to manage and grew faster on poor soils, securing the continuous supply of raw material for the growing timber industry (Broda, 2000). Such an environment is particularly dangerous for Poland's peatlands because monoculture tree plantations have simplified linkages in food webs and thus are more sensitive to fires, strong winds, droughts, and insect outbreaks (Chapin et al., 2012), which also poses a threat to peatlands. It should be strongly emphasized here that such extreme phenomena have become more common in recent years around the world (Seidl et al., 2014; Westerling, 2016). These negative impacts have been recorded for various peatlands, including those in Central and Eastern Europe (Leonardos et al., 2024; Łuców et al., 2021).*

**Line 73: also here I think it would be suitable to mention what kind of insect it was. Bark beetle?**

**Author's response**: Thank you to the reviewer for the comment. We agree that it is worth specifying that it is a *Panolis flammea* outbreak.

**Actions**: We corrected the sentence as the reviewer suggested, explaining that there was a Panolis flammea outbreak between 1922 and 1924.

**Before**: *These forests were affected by some of the most severe environmental disasters of the 20$^{th}$ century that took place in pine-dominated forests across Central and Eastern Europe – the 1922-1924 insect outbreak and the 1992 fire.*

**After**: *These forests were affected by some of the most severe environmental disasters of the 20$^{th}$ century that took place in pine-dominated forests across Central and Eastern Europe – the 1922-1924 Panolis flammea outbreak and the 1992 fire.*

**Line 76: please replace 'dramatic' with 'extreme'**

**Author's response**: Thank you to the reviewer for the comment. We agree that it is worth replacing 'dramatic' with 'extreme'.

**Actions**: We corrected the sentence as the reviewer suggested, replacing word 'dramatic' with 'extreme'.

**Before**: *However, not all the evidence of past dramatic events has been well preserved in the previously studied core [...]*

**After**: *However, the interpretation of these extreme events based solely on these two cores appears to leave many questions unanswered and highlights the need for further research into the impact of insect outbreaks and fires on peatland ecosystems.*

**Line 76 -78: This seems insufficiently explained. Did all three studies mentioned in line 75 use the same sediment core? Did they search for evidence but it was not found in the core, thus suggesting that evidence did not preserve?**

**Author's response**: We appreciate the reviewer's curiosity and request for detailed information about previous palaeoecological studies from the Noteć Forest. To date, only two cores from a single Rzecin peatland have been analyzed and the results coming from these two cores have been published in the referred publications. Barabach (2014) is a PhD thesis (monography published in Polish) focusing on palynological data, an article by Lamentowicz et al. (2015) focuses on the reconstruction of hydrological changes in Rzecin peatland, whereas Milecka et al. (2017) offers a summary of palaeoecological research undertaken at Rzecin peatland. However, these studies have left many questions unanswered, which motivated the continuation of the work on this topic and the writing of this manuscript.

**Actions**: We significantly rewrote sentences referring to previous studies, detailing how many cores were previously studied and why we decided to continue the research about the impact of ecological disasters on ecosystems in the Noteć Forest.

**Before**: *The only palaeoecological data documenting these events in the Noteć Forest come from the Rzecin peatland (Barabach, 2014; Lamentowicz et al., 2015; Milecka et al., 2017). However, not all the evidence of past dramatic extreme events has been well preserved in the previously studied core, leaving the question of the impact of insect outbreaks and fire on peatlands open for further investigation.*

**After**: *The only palaeoecological data documenting these events in the Noteć Forest were derived from two cores taken from the Rzecin peatland (Barabach, 2014; Lamentowicz et al., 2015; Milecka et al., 2017). However, the interpretation of these extreme events based solely on these two cores appears to leave many questions unanswered and highlights the need for further research into the impact of insect outbreaks and fires on peatland ecosystems.*

**Line 78 – 79: about small peatlands, seems out of place here. The text makes sense without referring to the size of peatlands here.**

**Author's response**: Thank you to the reviewer for the comment. We agree that this sentence should be removed from the introduction because it disrupts the text's structure.

**Actions**: We removed the sentence as the reviewer suggested.

**Before**: *[…] leaving the question of the impact of insect outbreaks and fire on peatlands open for further investigation. Small peatlands are usually less resilient to disturbances than large ones (Lamentowicz et al., 2008). The changes caused by extreme events can lead a peatland to reach a critical transition, that is, to cross a tipping point after which it does not return […]*

**After**: *[…] leaving the question of the impact of insect outbreaks and fire on peatlands open for further investigation. The changes caused by extreme events can lead a peatland to reach a critical transition, that is, to cross a tipping point after which it does not return […]*

**Methods:**
**Most of the methodology reads fine and is logical to me, but this being the first manuscript that I'm reading that studies Nd isotopes I did not get enough information about why and how it is used. Sometimes briefly referring to literature is not enough. It probably would suffice to borrow a few sentences from the publication by Marcisz et al (2023b).**

**Author's response**: Indeed, neodymium isotopes are still not commonly used in palaeoecological studies of peatlands. We agree that adding more context to the 'Methods' section will help the readers understand this method.

**Action**: We added a bit more context to neodymium methodology at the beginning of 'Neodymium isotopes' methodology chapter. We also added information about the sampling of neodymium reference samples in the 'Fieldwork and sampling' section as this information was missing in the previous version of the manuscript.

**Line 146: replace 'are' with 'were'.**

**Author's response**: We appreciate the reviewer's work on the grammatical correctness of the manuscript. We agree that the proposed sentence form is proper.

**Action**: We have corrected the sentence as the reviewer suggested.

**Before:** *[…] of plant macrofossils, which may signal changes in peat accumulation rates, are inputted using the Boundary command.*

**After:** *[…] of plant macrofossils, which may signal changes in peat accumulation rates, were inputted using the Boundary command.*

**Line 148: It makes sense that these two radiocarbon dates were rejected from the model, but I don't understand the explanation in the text.**

**Author's response**: We appreciate the reviewer's contribution to improving the manuscript and taking care of its logical content. We agree that the sentence was not clear and needed rewriting.

**Action**: We corrected the sentence, explaining that two dates were rejected because they were outside the main trajectory of the model.

**Before:** *Two dates (laboratory code – Poz-150636 and Poz-150390) were rejected because they were after the initial modelling trajectory of the model.*

**After:** *Two dates (laboratory code – Poz-150636 and Poz-150390) were rejected because they were outside the main trajectory of the model.*

**Line 182: Perhaps you could keep the same terminology here as for the previous part about the plant macrofossils.**

**Author's response**: In our opinion the style and terminology in which testate amoeba and plant macrofossil methods are explained is similar, therefore, we do not see what we can possibly change to improve it.

**Results:**
**Figure 2 looks very nice, but Figure 3 and Figure 4 could perhaps be a bit summarized and simplified (further). This is only an aesthetic opinion, though.**

**Author's response**: Thank you to the reviewer for the comment. Indeed, the font was too small. In addition, the charts could have been simplified. We agree with you.

**Actions:** We have corrected the figure by making the font larger. We have made the figure clearer. We also reduced the number of data presented by eliminating some of the dates from the timeline, as well as some of the values from the horizontal axes next to the curves.

In Figure 2, we made the font larger, and we removed part of the values from the horizontal axes and the age axis.

In Figure 3, we made the font larger, we removed non-discussed taxa curves (*Cirsium* – fruits), and some of the values from the horizontal and vertical axes.

In Figure 4, we made the font larger, we removed non-discussed taxa curves (*Salix*, *Populus*, *Tilia cordata*, *Abies alba*, *Acer*, *Pteridium aquilinum, Melampyrum*, *Urtica*, *Humulus/Cannabis*, *Centaurea cyanus*, *Fagopyrum esculentum* type, *Ambrosia artemisiifolia* type, Chenopodiaceae, *Plantago major*, Apiaceae undiff, *Potentilla* type, Brassicaceae, *Anthemis* type, *Aster* type, Cichoriaceae, *Galium* type, *Typha latifolia*, *Sparganium* type, *Drosera rotundifolia*, HdV-153 *Riccia*, Filicales monolete, *Scenedesmus*, *Spirogyra* type, *Mougeotia* type. *Pediastrum* – sum, HdV-128A, HdV-128B, HdV-1 *Gelasinospora* sp., HdV-30 *Helicoon pluriseptatum*, HdV-28 Copecoda spermatophores, HdV-31 *Archerella flavum*, HdV-32A *Assulina muscorum*, HdV-32B *Assulina semimulum*), and some of the values from the horizontal and vertical axes, and some of the values from the horizontal and vertical axes.

**Figure 2**
**Before***:*

[Figure]

**After**:

[Figure]

**Figure 3**
**Before:**

[Figure]

**After:**

[Figure]

**Before:**

[Figure]

**After:**

[Figure]

My biggest comment on the results is on the statistical analysis. I am quite skeptical about how a line was fitted through the data points (Fig 5), as the PCA values seem quite stable between 1775 – 1920. Have you already tried to plot the rate of change?

**Author's response:** Thanks for the thoughtful comment. While the PrC scores appear relatively stable during the 18[th] into the early 19[th] centuries, there is a subtle but consistent increasing trend beginning around the late 16[th] century (also identified by the CONISS in the pollen diagram). This upward trend contrasts with the previous period which shows higher short-term variability, reflecting fluctuations in the relative abundance of NPPs at this time, mainly cyanobacteria. This increase in the PrC scores becomes more pronounced during the 19[th] century as this taxon disappear from the record.

This trend is more clearly captured in the asGAM fit, shown in Supplementary Figure A1. However, adaptive spline GAMs cannot yet incorporate a temporal correlation structure, making them unsuitable for this analysis. In contrast, the GAMM used here includes a

correlation structure that penalizes sharp deviations between temporal adjacent samples, resulting in a smoother fit that is better able to detect sustained directional changes over time and to distinguish underlying shifts from random noise. In this case, it appears the model includes the small but consistent increase around 1775 AD as statistically significant in terms of slope, even if this is not visually dramatic in the raw PrC values.

Importantly, the purpose of the PrC analysis is to identify periods of significantly rapid rates of change (lines 266-267|). The GAMM framework, following the approach of Burge et al., 2023, allows for the identification of directional shifts and when these begin to accumulate beyond the level expected from random variation (lines 267-269). This is included in the text (lines 271- 273) and we have added additional clarification where needed, including in the figure caption.

**In the caption of Fig 3 it should be mentioned that DWT is calculated from the testate amoeba. And perhaps it should be mentioned again that the two largest recorded calamities are included again, like in Fig 2.**

**Author's response**: We thank the reviewer for his suggestions to improve the caption of Figure 3. We agree that this additional information is needed.

**Actions**: We completed the caption with the information that the reconstruction of water levels and pH was carried out based on testate amoeba analysis. We also added information about the marked periods of the analyzed ecological disasters in Figure 3.

**Before**: *Figure 3. A diagram showing macrofossil percentages, macroscopic charcoal concentrations and influx as a local fire proxy. Depth-to-water table and pH curves for 27–0 cm layers are also presented. Ten times exaggeration is marked.*

**After**: *Figure 3. A diagram showing macrofossil percentages, macroscopic charcoal concentrations and influx as a local fire proxy. Testate amoeba-based depth-to-water table and pH curves for 27–0 cm layers are also presented. The timing of the most critical catastrophic disasters in the 20th century is also marked. Ten times exaggeration is presented.*

**The second line in Fig 3, marking between zone 2 and zone 3, should be replaced with a thin one. It actually confused me.**

**Author's response:** All lines marking the transitions between zones in Figure 3 are of equal thickness - 0.8 pt. The entire figure was prepared in a single graphics program. We have no idea why the thicknesses of the lines seem different to the reviewer.

**Line 348: microscopic charcoal already starts to increase halfway the phase, it appears.**

**Author's response**: I appreciate the reviewer's input on the consistency of the text with the figures. We agree that the comment is right.

**Actions**: We corrected the sentence, taking into account the differences between the first and second halves of the zone.

**Before**: *Through much of the phase 2, fire activity is low.*

**After**: *For the first half of phase 2, fire activity is low, but increases in the second half.*

**Line 362: It is already clear from Fig 3 that you did not reconstruct DWT up until zone 4.**

**Author's response**: We agree that the information is already known from Figure 3. On the other hand, we would like Figure 3 to be completely consistent with the text. We would also like the text to clearly indicate the reasons why we decided to hide the reconstruction of the water level up to zone 4 in Figure 3.

**Line 367: this is not a result but rather a point of discussion, or conclusion.**

**Author's response:** We thank the reviewer for his comment. We agree with the reviewer that the sentence was an interpretation or conclusion.

**Actions**: We removed the sentence from this section as the reviewer suggested.

**Line 413 – 414: this is about zone 4 and not zone 5.**

**Author's response**: We thank the reviewer for his comment and care about the logical structure of the text. We agree that there was an error in phase numbering.

**Actions**: We have corrected the numbering of the zones in the sentence.

**Before**: *G. discoides dominates for most of the phase 4, and the abundance of N. tincta increases towards its end.*

**After**: *G. discoides dominates for most of the phase 5, and the abundance of N. tincta increases towards its end.*

**My suggestion is to move Table 2 to the supplementary materials, since values are already displayed in Fig 2.**

We agree with the reviewer that the neodymium isotope values can be seen in Fig. 2. However, they refer only to peat samples, not to surface samples. The table containing the values of peat samples and surface samples allows the neodymium isotope signatures to be compiled in one place, and additionally allows the values to be compared between sample types. In our opinion, a better solution is to leave the table in the text.

**Discussion**
**I'm missing a discussion about why the fluctuating values in the PCA don't appear to reflect any significant environmental changes. So what caused these fluctuations? Is the pattern unique for this record or was it found in other studies as well?**

**Author's response:** Thank you for your thoughtful comments. We suspect there may be some confusion arising from the conflation of Principal Correspondance Analysis (PCA) and Principal Response Curves (PrC), the latter being the method used in this study. While both are ordination techniques used to summarise variation in multivariate datasets, they serve different purposes. PrCs explore temporal changes relative to a baseline or control- in this case, the first sample in the dataset. The differences in the PrC curves reflect how the pollen spectra deviate

over time from this point- based on shifts in the relative abundance of both pollen and non-pollen palynomorphs in the record.

The trend in the PrC aligns broadly with the patterns seen in the data, as shown by the correspondence between the PrC scores and the relative contributions of deciduous trees, arboreal pollen, *Pinus sylvestris* type and NPPs. While these fluctuations are driven by changes in the palynological record, the primary aim of the PrC was not to characterise these trends (as one might do using a PCA) but to track the rate of change over time by fitting a GAMM to the PrC axis against time.

The shifts in pollen and NPP are well discussed in the text, so to further clarify the distinction between PrC and other, more commonly used ordination methods, we have added a statement to line # stating that the PrC results trace changes in the relative abundance of pollen and NPP over time and have added further clarification of the purpose of the PrC in the methods section.

**L 488: I'm sure the fitting of the complete function was statistically significant, but the portion 1775 – 1900 seems to be deviating from the data points.**

**Author's response:** This issue is (somewhat) acknowledged and referenced in the main text (line 468 - 474, Supplementary Figure A1). In brief, although we tested different numbers for k, the flexibility of the model fit remained unchanged, constrained by the correlation structure used within the model, which is essential to account for temporal autocorrelation in the data (see Burge et al., 2023- and Simpson 2018).

As shown in Supplementary Figure A1, an adaptive spline GAM (asGAM) produced a better visual fit to the PrC scores. However, these cannot currently be used within the GAMM framework, which is required to specify correlation structures (See Simpson, 2018 and Burge et al., 2023). Our focus is on the slope of the fitted line- i.e. the rate of change, rather than the individual points. While the slope from the asGAM increases more sharply closer to the period associated with the infestation (so supports are argument better!), the selected model still captures the key shift, albeit earlier. Given current methodical limitations with using asGAMs in this framework, we opted for the more conservative estimate provided by the GAMM.

We have clarified the summary of the model performance in the results (lines 476-482), mirroring the above information.

**L 500 – 501: those are quite lot of references just to mention that this is the first record that captures a much longer history of the Notec Forest.**

**Author's response**: Thank you for this comment. However, we propose to leave all these citations as they highlight the lack of data from the Noteć Forest. All these references refer to a single site - the Rzecin bog. To date, no other work has been produced from other peatlands. We want to emphasize this, so we kept all these citations.

**L 502 – 505: This sentence was quite confusing to me. Do you mean to say, that the analysis of your record allow for the reconstruction of not only the forest but also the peatland itself, and how it reacted to forestry practices?**

**Author's response**: Thank you to the reviewer for the comment. Indeed, the sentence was constructed imprecisely and was unclear.

**Actions**: We rewrote the sentence to make it more logical.

**Before**: *The knowledge of the historical background is essential for the interpretation of the ecosystem response to forestry practices because it enables tracing not only the composition of the forest surrounding the peatland but also the peatlands' hydrological and trophic conditions (Bąk et al., 2024).*

**After**: *The knowledge of the historical background is essential for the interpretation of the complex response of the peatland ecosystem to a change in forest management, as it allows for the long-term tracing of reference conditions relating to both the composition of the forest and the trophic and hydrological variants of the peatland (Bąk et al., 2024).*

**L 512: I think you meant replaced, rather than misplaced**

**Author's response**: We thank the reviewer for his contribution to the grammatical correctness of the manuscript. However, we do not use the verb 'misplace' but 'displace', which is actually a synonym for the verb 'replace'. In the University of Cambridge's online dictionary, 'displace' means 'to replace something or someone'. So, we stayed with the version with the verb 'displace'.

**L 530 – 532: You only investigated the last 2000 year of the forest.**

**Author's response**: Yes, that is correct, here we reconstructed last 2000 years of environmental history of the site. We guess that what the reviewer is referring to here is that young glacial sediments are older than 2000 years and so the connection between neodymium data and our record may be weak. Neodymium isotopes are stable isotopes therefore they do not change their signatures through time. Thanks to that, by comparing Nd ratios obtained from the peat core and the soil in peatland's surrounding, we can define whether the mineral supply (and also recorded disturbances) was of local or extra-local origin.

**L547 – 550: See my previous comments on Fig 5 and L 488.**

**Authors response:** I guess see my previous response to previous comments. However, I would suggest changing the text here from critical transitions to 'periods of significantly rapid change' or words to that effect- because that is what the GAMM model shows. It's not necessarily wrong that they can reflect critical transitions, but the model doesn't specifically identify critical transitions like other methods can, e.g. TITAN it can detect non-critical transitions too if they are happen quickly enough!).

**L633: Have you tried to isolate butterfly wing scales (Girona, 2018)?**

**Author's response**: We thank the reviewer for his comment. Unfortunately, we did not use the method by Girona et al. (2018) to extract the wings of the insect.

**Actions**: We have added information to this section of the text that there are methods for extracting butterfly wings, but they were not the subject of our analysis. We have rewritten the text accordingly.

**Before**: *Unfortunately, they do not preserve well in the sediment (Bąk et al., 2024).*

**After**: *Unfortunately, they do not preserve well in the sediments (Bąk et al., 2024). However, we emphasize that we did not use advanced extraction methods the delicate structures of the butterfly wing remains (Montoro Girona et al., 2018), but only observation under light and stereoscopic microscopes when viewing the samples in the analyses used.*

**L690: what are stable ecosystem links, exactly?**

**Author's response**: Thank you to the reviewer for the comment. Indeed, the sentence was constructed imprecisely and was unclear. It is about simplified linkages in the food web, which result in less resilience of the ecosystem to various types of disturbances.

**Actions**: We corrected the sentence's meaning, clarifying what linkages we meant. We hope that after the correction, this part of the text is no longer in doubt.

**Before**: *Fire danger is also a result of the young age of the tree stands, which have not yet developed stable ecosystem links.*

**After**: *Fire danger is also a result of the young age of the tree stands, which have not yet developed stable ecosystem links in food webs.*

**L703: are these bark beetle outbreaks? And this was not evident in your record?**

**Author's response**: Thanks to the reviewer for the comment. We would like to remind you that we are describing the *Panolis flammea* outbreak, not the bark beetle outbreak. After the *Panolis flammea* outbreak in 1922-1924, the Noteć Forest was affected by many other insect outbreaks. However, they were smaller, less severe and involved different locations of the Forest. They did not occur near the study site. So, it is difficult to get a clear signal of these events.

**Action**: We have rewritten the text, adding information about later known pest outbreaks. We also explained why the record of these other outbreaks is invisible in our data.

**Before:** *In the 1970s, Hernik (1979) and Ratajszczak (1979) signalled that the tree stands of the Noteć Forest were weakened by repeated insect outbreaks.*

**After:** *In the 1970s, Hernik (1979) and Ratajszczak (1979) signalled that the tree stands of the Noteć Forest were weakened by repeated insect outbreaks (Panolis flammea: 1956; Lymantria monacha: 1947, 1964; Barbitistes constrictus: 1964; Diprion pini: 1961; Bupalus piniarius, 1966; Dendrolimus pini: 1970). Compared to the 1922-1924 Panolis flammea outbreak, however, they were smaller, less severe, and covered different locations of the Noteć Forest, rather than a larger area.*

**L725: It could also have played a role that Mialy is situated next to a railroad and your study site was surrounded by trees.**

**Author's response**: Thank you for the reviewer's comment. We clarify that the fire started near the railroad north of the village of Miały (Fig. 6). It spread very quickly in the eastern direction, destroying a dense forest complex. The fire also directly reached our site located in this dense forest complex, as we show in Fig. 6. It seems that the location of the research site, a few kilometers from the railroad, is of little importance for these considerations, since the fire directly

**Action**: We have modified and improved the text of this paragraph. Previously, the description might have raised doubts about whether the fire reached the studied peatland or not. We hope that the description is now clear and logical.

**Before:** *[…] these values do not reflect the actual scale of the forest destruction, especially since the fire took place near the peatland (Fig. 5). A smaller-than-expected signal from the 1992 fire in charcoal analysis was also obtained by Barabach (2014) in the nearby Rzecin peatland. The small amount of macroscopic charcoal may be explained by the fact that the more intense the fire, the smaller the charcoal particles it produces (Schaefer, 1973). Additionally, before the particles are deposited, their dispersion by wind and water plays an important role (Patterson et al., 1987). By the time the fire reached the peatland, heavy rain had fallen, reaching a value of 31.5 mm (Institute of Meteorology and Water Management, 2025). This rain stopped the smoke from spreading further away, however, it reached the Miały peatland (Fig. 6).*

**After:** *[…] these values do not reflect the actual scale of the forest destruction, especially since the fire also took place on the peatland (Fig. 6). A smaller-than-expected signal from the 1992 fire in charcoal analysis was also obtained by Barabach (2014) in the nearby Rzecin peatland. The small amount of macroscopic charcoal may be explained by the fact that the more intense the fire, the smaller the charcoal particles it produces (Schaefer, 1973). Additionally, before the particles are deposited, their dispersion by wind and water plays an important role (Patterson et al., 1987). Shortly after the fire reached the peatland, heavy rain had fallen, reaching a value of 31.5 mm (Institute of Meteorology and Water Management, 2025). This rain stopped the fire from spreading further away and significantly limited the movement of charcoal by the wind.*

**L756: How is slow tree growth affecting forest fires?**

**Author's response**: We would like to thank the reviewer for his comment. We realize that this section of the text may have been questionable. The idea was that frequent fires in May are a consequence of the lack of rainfall in May. In another study, we observed that the growth of pine annual rings from another monoculture complex in Poland showed a negative correlation with the amount of precipitation in May. This observation strengthens the message about the high number of fires in May resulting from the drought in that month.

**Action**: We have rewritten this section of the text so that it is more logical and not questionable.

**Before:** *Forest fires in Poland were, therefore, frequent but covered small areas (0.4 ha/fire on average). Most fires in Poland occurred in May (more than 25%). This pattern is vital when compared with dendroclimatic data. A recent study from the pine-dominated Tuchola Forest in*

*Poland revealed a negative correlation between Scots pine growth and rainfall in May (Bąk et al., 2024). A water deficit in May carries, therefore, many dangerous consequences.*

**After:** *Therefore, forest fires in Poland were-frequent but covered small areas (0.4 ha/fire on average). Most of the fires in Poland occurred in May (more than 25%), a significant percentage of which were drought-induced. A recent study from the pine-dominated Tuchola Forest in Poland revealed a negative correlation between Scots pine growth and rainfall in May (Bąk et al., 2024), which indeed indicates a water deficit in that month.*

---

## Author Response (AR2)

**Responses to review No. 1**

**Reviewer general comment:** The authors have sufficiently answered all my questions and made the suggested changes. Thanks to the authors for their efforts. I have a few minor suggestions below. All of these line numbers correspond to the version with track changes.

**Author's response**: We sincerely thank the reviewer for taking the time to reread the manuscript, for the valuable comments, and for recognizing the value of our work. Below, we provide detailed, point-by-point responses to all remarks and suggestions included in the review.

**Line 291: sentence suggestion: "However, REML was used to reanalyse the data in place of ML as a smoothing parameter, although it did not make an appreciable difference to the results."**

**Author's response**: The correction proposed by the reviewer indeed improves the sentence structure, making it stylistically better and more coherent.

**Actions**: We have adjusted the sentence as suggested by the reviewer.

**Before**: However, REML was used to reanalyse the data in place of ML as a smoothing parameter, and it didn't make an appreciable difference to the results.

**After**: However, REML was used to reanalyse the data in place of ML as a smoothing parameter, although it did not make an appreciable difference to the results.

**Line 528: Pinus sylvestris type should be italicized.**

**Author's response**: We appreciate the reviewer's attention to this detail in the manuscript.

**Actions**: We have adjusted the sentence as suggested by the reviewer.

**Before**: […] the PrC scores and the relative contributions of deciduous trees, arboreal pollen, Pinus sylvestris type and NPPs.

**After**: […] the PrC scores and the relative contributions of deciduous trees, arboreal pollen, *Pinus sylvestris* type and NPPs.

**Lines 535-539: Sorry to change the sentence again. Suggest saying: "Previous multi-proxy palaeoecological studies exist from the Noteć Forest; however, they were unable to…". Also, can you clarify what you mean by "provide such information?" What information were they unable to provide, specifically? I assume the admixture of other species.**

**Author's response**: The correction proposed by the reviewer indeed improves the sentence structure, making it stylistically better and more coherent. We acknowledge that greater precision was needed in specifying the scope of the information presented.

**Actions**: We have adjusted the sentence as suggested by the reviewer. We have expanded the sentence to make it clearer and to eliminate any potential ambiguity.

**Before**: Previous multi-proxy palaeoecological studies exist from the Noteć Forest; however, those previous ones were unable to provide such information because the cores collected from the Rzecin peatland […]

**After**: Previous multi-proxy palaeoecological studies from the Noteć Forest exist; however, they were unable to provide information on the proportion of admixture tree species in the forest composition prior to the onset of planned forest management. The cores collected from the Rzecin peatland […].

**Line 588: sentence suggestion: "The instability of the ecosystem witnessed after the 20th century is a consequence of…."**

**Author's response**: We appreciate the suggestion, which improves the sentence's style and clarity.

**Actions**: We have adjusted the sentence as suggested by the reviewer.

**Before***: The instability of the ecosystem is a consequence of the introduction of planned forest management […]

**After**: The instability of the ecosystem witnessed after the 20$^{th}$ century is a consequence of the introduction of planned forest management […]

**Lines 685-686. I believe there are some typos or maybe missing words in this sentence. Should it read, "However, we emphasize that we did not use advanced extraction methods that could potentially preserve the delicate structures of the butterfly wing remains…."?**

**Author's response**: We appreciate the suggestion, which improves the sentence's style and clarity.

**Actions:** We have adjusted the sentence as suggested by the reviewer.

**Before***: However, we emphasize that we did not use advanced extraction methods the delicate structures of the butterfly wing remains […]

**After**: However, we emphasize that we did not use advanced extraction methods that could potentially preserve the delicate structures of the butterfly wing remains […]

**Line 718: "We noted a gradual transition from a moderately rich fen to a poor fen in phase 4…"**

**Author's response**: We thank the reviewer for their attention to the grammatical accuracy of the manuscript.

**Actions:** We have corrected the sentence as suggested by the reviewer.

**Before**: We noted a gradual transition from the moderately rich fen to the poor fen in phase 4 (ca. 1660-1960 cal. CE**)**.

**After**: We noted a gradual transition from a moderately rich fen to a poor fen in phase 4 (ca. 1660-1960 cal. CE**)**.

**Lines 721-722: "Bąk et al. (2024) pointed out that such changes are a result of…"**

**Author's response**: We thank the reviewer for their attention to the stylistic accuracy of the manuscript.

**Actions**: We have corrected the sentence as suggested by the reviewer.

**Before**: Bąk et al. (2024) pointed out that such changes are characteristic as a result of forest management activities and can be caused by drainage and transformation in forest species composition.

**After**: Bąk et al. (2024) pointed out that such changes are a result of forest management activities and can be caused by drainage and transformation in forest species composition.

**Line 727: "…documented by the complete disappearance of Cyanobacteria and algae…"**

**Author's response**: We thank the reviewer for their attention to the stylistic accuracy of the manuscript.

**Actions**: We have adjusted the sentence as suggested by the reviewer.

**Before:** The change in trophic conditions at this time, and the concomitant change in hydrological conditions, are also documented by the completely disappearing Cyanobacteria and algae […]

**After:** The change in trophic conditions at this time, and the concomitant change in hydrological conditions, are also documented by the complete disappearance of Cyanobacteria and algae […]

**Line 730: "…suggesting a lowering of the water table and substantial water table fluctuations…"**

**Author's response:** We thank the reviewer for their attention to the grammatical accuracy of the manuscript.

**Action:** We have corrected the sentence as suggested by the reviewer.

**Before:** […] species that tolerate unstable hydrological conditions became dominant, suggesting the lowering of the water table and substantial water table fluctuations […]

**After:** […] species that tolerate unstable hydrological conditions became dominant, suggesting a lowering of the water table and substantial water table fluctuations […]

**Line 743: "… in a peatland…" Which peatland are you referring to?**

**Author's response**: We agree with the reviewer that the sentence requires greater precision.

**Action**: We have adjusted the sentence as suggested by the reviewer.

**Before:** In the period of the transition of trophic and hydrological conditions in a peatland (ca. 1925-1960 CE) […]

**After:** In the period of the transition of trophic and hydrological conditions in a Miały peatland (ca. 1925-1960 CE) […]

**Line 761. Sphagnum should be italicized.**

**Author's response**: We appreciate the reviewer's attention to this detail in the manuscript.

**Action**: We have corrected the sentence as suggested by the reviewer.

**Before:** […] decided to separate the developmental phase of the object they studied, which they referred to as Sphagnum bog […]

**After:** […] decided to separate the developmental phase of the object they studied, which they referred to as *Sphagnum* bog […]

**Responses to the Editor's Comments**

**Reviewer general comment:** Thank you for the submitted revision. Apart of minor changes suggested by one reviewer, I additionally found minor issues that should be addressed

**Author's response**: We thank the editor for the comments on the manuscript provided below. We agree that these issues required clarification, revision, or elaboration. We have made every effort to revise the text following the editor's expectations.

**Fig. 4 - 'Charcoals' change to 'charcoal'**
**What are the white bands around 20cm?**

**Author's response**: Thank you for the comment. We agree that the figure needs to be improved in this regard and that the presence of the white bands should be properly explained.

**Action**: We have corrected the error by changing "charcoals" to "charcoal." The white bands visible in the figure have been explained in the methodology section related to the palynological analysis — two samples (from depths 19–18 cm and 17–16 cm) were excluded from the analysis due to extremely low pollen concentration. We have updated the figure caption to include this information.

**Before:**

[Figure]

Figure 4. Pollen diagram with selected taxa presented (full list of taxa is provided in the associated open dataset). Pollen percentages are shown in black, and 10 times exaggeration is marked. Microscopic charcoal concentrations and influx as an extra-local fire proxy are also presented.

**After:**

[Figure]

Figure 4. Pollen diagram with selected taxa presented (full list of taxa is provided in the associated open dataset). Pollen percentages are shown in black, and 10 times exaggeration is marked. Microscopic charcoal concentrations and influx as an extra-local fire proxy are also presented. Two samples (depths: 19–18 and 17–16 cm) were excluded from the diagram due to extremely low pollen concentration (no data shown for these depths).

**L. 283—First, you say that palynological data were used to calculate PrC. Here, you write that NPPs also contributed to the inferred change (which I think is a wrong approach). Could you clarify?**

**Author's response**: The term *palynological* does not exclusively refer to pollen; it encompasses all palynomorphs, including NPPs. Therefore, there is no contradiction in our use of the term here.

The reviewer does not explain why they consider our approach inappropriate, but we interpret their concern as relating to the use of proxies that reflect environmental changes at different spatial scales. While most RoC (Rate-of-Change) studies rely on a single proxy, the combination of multiple proxies, including pollen and algae, has been applied in previous research (e.g., Abrook et al., 2019). Theoretically, there is no methodological issue with combining proxies in RoC analyses, provided that the approach is clearly justified.

By integrating pollen and NPPs, we aimed to provide a more nuanced understanding of compositional change resulting from hydrological and trophic shifts by using the full palynological dataset. Our rationale was that we intended to detect vegetation shifts across both local and broader landscape scales. In practice, however, the pollen diagram alone shows limited variation until relatively recently, the dominant signal originates from the NPPs, highlighting the primarily local character of these environmental changes. We chose to present the composite signal, as it offered the clearest and most comprehensive picture of change through time, although using the NPPs alone would have led to the same interpretation.

**Action**: We revised the methodological description of the statistical analysis to clarify that we examined not only the pollen record (reflecting changes in vegetation composition at both regional and local scales) but also the NPP record (reflecting local trophic and hydrological changes on the peatland). We also added that combining multiple proxy indicators in Rate-of-Change analyses has already been successfully applied in previous scientific studies.

We also revised a section of the discussion to clarify that the PrC analysis reveals not only rapid changes in forest composition but, even more importantly, changes in vegetation, hydrological, and trophic conditions on the peatland. These changes resulted from forest management practices and subsequent catastrophic events to which such management likely contributed.

We also removed the section of the discussion related to the fire that referred to the PrC analysis, as it did not provide any additional insights beyond what had already been discussed earlier.

**Before**: To quantify periods of rapid botanical change and recovery, we apply the principal response curves (PrC) to the data, as outlined by Burge et al. (2023) in their R package 'baselines'.

**After**: To quantify periods of rapid vegetation change in the forest (regional scale) and on the peatland (local scale), as well as hydrological, and trophic shifts on the peatland, we apply the principal response curves (PrC) to the data, as outlined by Burge et al. (2023) in their R package 'baselines'.

*Lines 267-270 in the new file with tracking changes.*

**Before**: Thus, PrC results trace changes in the relative abundance of pollen and NPP over time. This method is useful for detecting changes in data with a strong underlying gradient in palaeoecological studies (Van Den Brink and Ter Braak, 1999; De'ath, 1999).

**After**: Thus, PrC results trace changes in the relative abundance of pollen and NPP over time. While most RoC (Rate-of-Change) studies rely on a single proxy, the combination of multiple proxies, including pollen and algae, has been applied in previous research (e.g., Abrook et al., 2020).
This method is useful for detecting changes in data with a strong underlying gradient in palaeoecological studies (Van Den Brink and Ter Braak, 1999; De'ath, 1999).

*Lines 273-276 in the new file with tracking changes.*

**Before**: Around this time, the PrC analysis began to reveal periods of significant and rapid change in the palynological record. Consequently, the forest has continued to undergo substantially rapid changes ever since, unlike the preceding changes.

**After**: Around this time, the PrC analysis began to reveal periods of significant and rapid change in the palynological record. Since then, both the forest and, even more so, the Miały peatland within it have continued to undergo substantially rapid changes ever since, unlike the preceding changes.

*Lines 515-518 in the file with tracking changes.*

**Before**: Only an enclave of several hectares of deciduous old-growth forest resisted the fire. This event roughly coincides with the period of substantial rapid change identified by the PrC curve (Fig. 5), suggesting that this change may have contributed to the rapid alteration of the forest ecosystem reflected in pollen record.

**After**: Only an enclave of several hectares of deciduous old-growth forest resisted the fire.

*Lines 753-757 in the new file with tracking changes.*

**L. 298 - What kind of constrained ordination?**
**Where is the significance test of PrC, which you mention in the discussion?**

**Author's response**: We used Principal Response Curves (PrC), a constrained ordination technique used to assess time series data in ecological studies, which is explained in the text above.

The methodological foundations of the statistical analysis are described in detail in the "Statistical analyses" section.

*Lines 267-294 in the new file with tracking changes.*

In response to the question regarding the significance test of PrC: This is not strictly a test of the PrC but a test of significantly rapid rates of change based on the GAMM fitted to the PrC data, as explained in the section 'statistical methods'- lines 288 to 289 (in the tracking change file): "*Periods of significant change were identified in the GAMM models by calculating the time intervals, where the confidence intervals surrounding the first derivative did not include zero.*

**Action:** We made sure that wherever references to statistically significant rates of change appear in the text, the correct terminology is used (i.e., rate-of-change analysis, significant rate of change, etc.). Where necessary, we made the appropriate changes.

**Before:** The results of the PrC analysis proved to be statistically significant, confirming the occurrence of critical transitions in the peatland on a scale that was not observed in the older part of the core.

**After:** The results of the PrC analysis indicated a significant rate of change, confirming the occurrence of critical transitions in the peatland on a scale that was not observed in the older part of the core.

*Lines 519-521 in the new file with tracking changes.*

**Before:** Nevertheless, all three above-mentioned disturbance factors (introduction of planned forest management, 1922-1924 outbreak, and 1992 fire) affected the condition of the peatland and were recorded as statistically significant critical transitions in the GAMM model (Fig. 5).

**After:** Nevertheless, all three above-mentioned disturbance factors (introduction of planned forest management, 1922-1924 outbreak, and 1992 fire) affected the condition of the peatland and were recorded as significant rates of change in the GAMM model, which can be interpreted as critical transitions (Fig. 5).

*Lines 565-568 in the new file with tracking changes.*